# C-terminal calcium binding of α-synuclein modulates synaptic vesicle interaction

Janin Lautenschläger[1], Amberley D. Stephens [1], Giuliana Fusco[2,3], Florian Ströhl [1], Nathan Curry[1], Maria Zacharopoulou[1], Claire H. Michel[1], Romain Laine[1], Nadezhda Nespovitaya[1], Marcus Fantham[1], Dorothea Pinotsi[1,8], Wagner Zago[4], Paul Fraser[5], Anurag Tandon[5], Peter St George-Hyslop [5,6], Eric Rees[1], Jonathan J. Phillips [7], Alfonso De Simone[3], Clemens F. Kaminski [1] & Gabriele S. Kaminski Schierle [1]

Alpha-synuclein is known to bind to small unilamellar vesicles (SUVs) via its N terminus, which forms an amphipathic alpha-helix upon membrane interaction. Here we show that calcium binds to the C terminus of alpha-synuclein, therewith increasing its lipid-binding capacity. Using CEST-NMR, we reveal that alpha-synuclein interacts with isolated synaptic vesicles with two regions, the N terminus, already known from studies on SUVs, and additionally via its C terminus, which is regulated by the binding of calcium. Indeed, dSTORM on synaptosomes shows that calcium mediates the localization of alpha-synuclein at the pre-synaptic terminal, and an imbalance in calcium or alpha-synuclein can cause synaptic vesicle clustering, as seen ex vivo and in vitro. This study provides a new view on the binding of alpha-synuclein to synaptic vesicles, which might also affect our understanding of synucleinopathies.

[1] Department of Chemical Engineering and Biotechnology, University of Cambridge, West Cambridge Site, Philippa Fawcett Drive, Cambridge, CB3 0AS, UK. [2] Department of Chemistry, University of Cambridge, Lensfield Road, Cambridge, CB2 1EW, UK. [3] Department of Life Sciences, Imperial College London, London, SW7 2AZ, UK. [4] Prothena Biosciences Inc, South San Francisco, CA 94080, USA. [5] Tanz Centre for Research in Neurodegenerative Diseases, University of Toronto, Toronto, ON M5T 2S8, Canada. [6] Department of Clinical Neurosciences, Cambridge Institute for Medical Research, University of Cambridge, Cambridge, CB2 0XY, UK. [7] Department of Biosciences, Living Systems Institute, University of Exeter, Exeter, EX4 4QD, UK. [8] Present address: Scientific Center for Optical and Electron Microscopy, ETH Zurich, Otto-Stern Weg 3, CH8093 Zurich, Switzerland. Janin Lautenschläger and Amberley D. Stephens contributed equally to this work. Correspondence and requests for materials should be addressed to G.S.K.S. (email: gsk20@cam.ac.uk)

Alpha-synuclein is a 140-residue protein, which constitutes three major protein regions, the N terminus (aa 1–60), the non-amyloid-β component (NAC) region (aa 61–95), designated as the aggregation-prone region, and the C terminus (aa 96–140). Alpha-synuclein is localized at the pre-synaptic terminals[1], and, while structurally disordered in solution[2,3], it also exists in a partially structured, membrane-bound form. Indeed, alpha-synuclein can bind a variety of synthetic vesicles but displays a preference to bind to small, highly curved synthetic vesicles via its N terminus[4–10]. NMR studies of alpha-synuclein binding to synaptic-like synthetic vesicles have shown that this interaction is primarily triggered by the N-terminal residues, but interactions propagate up to residue 98, with the central region of the protein (residues 65–97) having a key role in modulating the binding affinity to the membrane[11] and in promoting the clustering of synaptic vesicles[12].

Moreover, although it has been shown that the N terminus of alpha-synuclein strongly interacts with lipid vesicles, it is important to note that so far all research on alpha-synuclein–lipid interactions has been carried out on synthetic lipid vesicles. It thus has yet to be shown how alpha-synuclein interacts with physiological synaptic vesicles which are clearly distinct from just lipid vesicles[13].

We hypothesized that calcium has a role in the normal physiological function of alpha-synuclein as alpha-synuclein is primarily localized at the pre-synaptic terminals where high calcium fluctuations occur, ranging up to hundreds of μM[14,15], and since calcium has been previously shown to bind to alpha-synuclein at its C terminus[16]. In addition, it is not clear what the calcium affinity to alpha-synuclein is, whether the C terminus is equally amenable to cations in the presence of synaptic vesicles, and how exposure to calcium would interfere with the synaptic vesicle binding capacity of alpha-synuclein. To answer these questions, we investigated firstly the calcium-binding properties of alpha-synuclein by NMR and mass spectrometry (MS). We then explored whether and how neutralization of negative charges on the C terminus impacts on the interaction of alpha-synuclein with lipids and synaptic vesicles. And finally, we tested whether the interaction of alpha-synuclein with synaptic vesicles impacts on synaptic vesicle homeostasis and on alpha-synuclein aggregation and toxicity related to Parkinson's disease (PD).

We show here that calcium interacts with the negatively charged C terminus of alpha-synuclein, having a $K_D$ in the range of 21 μM. Using synaptic vesicles isolated from rat brain, we performed chemical exchange saturation transfer (CEST) experiments in solution-state NMR, and show that the C terminus of alpha-synuclein has an increased tendency to interact with synaptic vesicles upon calcium binding. In the presence of calcium, alpha-synuclein exhibited specific clustering at the pre-synaptic terminal, which could be reversed by the addition of a calcium chelator. In contrast, VAMP2, a synaptic vesicle marker, was not affected by these calcium changes. These findings suggest that the normal physiological role of alpha-synuclein is to act as a calcium-dependent modulator of vesicle homeostasis at the pre-synaptic terminal. Using ventral mesencephalic neurons, we further show that treatment with dopamine, a neurotransmitter relevant in PD pathology, promoted the clustering of alpha-synuclein-positive vesicles. The latter was prevented by treatment with isradipine, a voltage-gated calcium channel inhibitor. Furthermore, lowering either the levels of alpha-synuclein or calcium prevented dopamine toxicity, indicating that both alpha-synuclein and calcium levels need to be finely balanced. This study provides a new view on the binding of alpha-synuclein to synaptic vesicles, which might also affect our understanding of synucleinopathies.

## Results

**Calcium increases the lipid binding of alpha-synuclein.** We recorded $^1$H-$^{15}$N heteronuclear single quantum correlation (HSQC) spectra of alpha-synuclein in solution NMR as a function of calcium concentration to determine the thermodynamics and structural nature of the calcium-binding mechanism. An analysis of the spectra identified the C terminus as the primary segment hosting chemical shift perturbations due to calcium binding. In addition, we found peak broadening for some residues in the NAC-region. Residues whose chemical shifts of the backbone amide N-H were mostly affected are aa 104, 107, 112, 119, 123, 124, 126, 127, 129, 130, 135, 136, 137 (Fig. 1a; Supplementary Fig. 1). To obtain information on the thermodynamics of the calcium affinity to alpha-synuclein, and as the number of calcium cations that are bound to alpha-synuclein was not known, we performed a global analysis by fitting the chemical shift perturbation using a multiple ligand model[17], which allows to estimate the binding constant when the exact number of ligands is not known. Similar models have also been used before to analyze multivalent interactions of other amyloidogenic proteins[18]. By performing best fit analysis, we obtained a $K_D$ of calcium affinity of 21 μM with the number of ligand cations, $L$, being 7.8 (Fig. 1b).

To provide further support on the number of ligand cations being bound to alpha-synuclein we performed electrospray ionization MS and found that at least six cations are bound to alpha-synuclein (Fig. 1c; Supplementary Table 1). We thus confirmed by two independent measurements that around 6–8 calcium ions can be bound to alpha-synuclein providing further support of our determined $K_D$. A dissociation constant of 21 μM lies well within the range of physiological pre-synaptic calcium fluctuations, reaching up to hundreds of μM in healthy neurons upon neuronal stimulation[14,15].

We then hypothesized that a neutralization of negative charges on residues at the C terminus via dynamic binding of positively charged calcium ions facilitates the interaction of alpha-synuclein with phospholipid membranes. To test this hypothesis, we incubated lipid Folch extracts from bovine brain with alpha-synuclein in the presence of calcium and other ions, including potassium, sodium, and magnesium and measured the interaction of alpha-synuclein with lipids via lipid pull down. In the presence of calcium, the amount of alpha-synuclein that precipitated with the lipids increased about fivefold. Magnesium and a high concentration of sodium ions also increased the amount of alpha-synuclein pulled down, but not to the same extent as calcium (Fig. 1d; Supplementary Fig. 2). A phase partitioning assay further established that the hydrophobicity of alpha-synuclein increases upon calcium addition, which is manifested by a higher abundance of the protein in the lipophilic detergent phase (Supplementary Fig. 3).

**The C terminus of alpha-synuclein binds to synaptic vesicles.** Having seen that calcium influences the binding of alpha-synuclein to lipids, we investigated the interaction of alpha-synuclein with synaptic vesicles isolated from rat brain and its dependence on calcium. Using measurements of chemical exchange saturation transfer (CEST) in solution NMR[11,12], we probed the binding affinity along the protein sequence. This analysis revealed that in the absence of calcium the interaction with synaptic vesicles was strongest via the N terminus. In the presence of calcium, however, the interaction of the C terminus and also for some residues of the NAC-region was increased (Fig. 2a), providing further insight into how calcium can modify the C terminus of alpha-synuclein and thereby induce a stronger

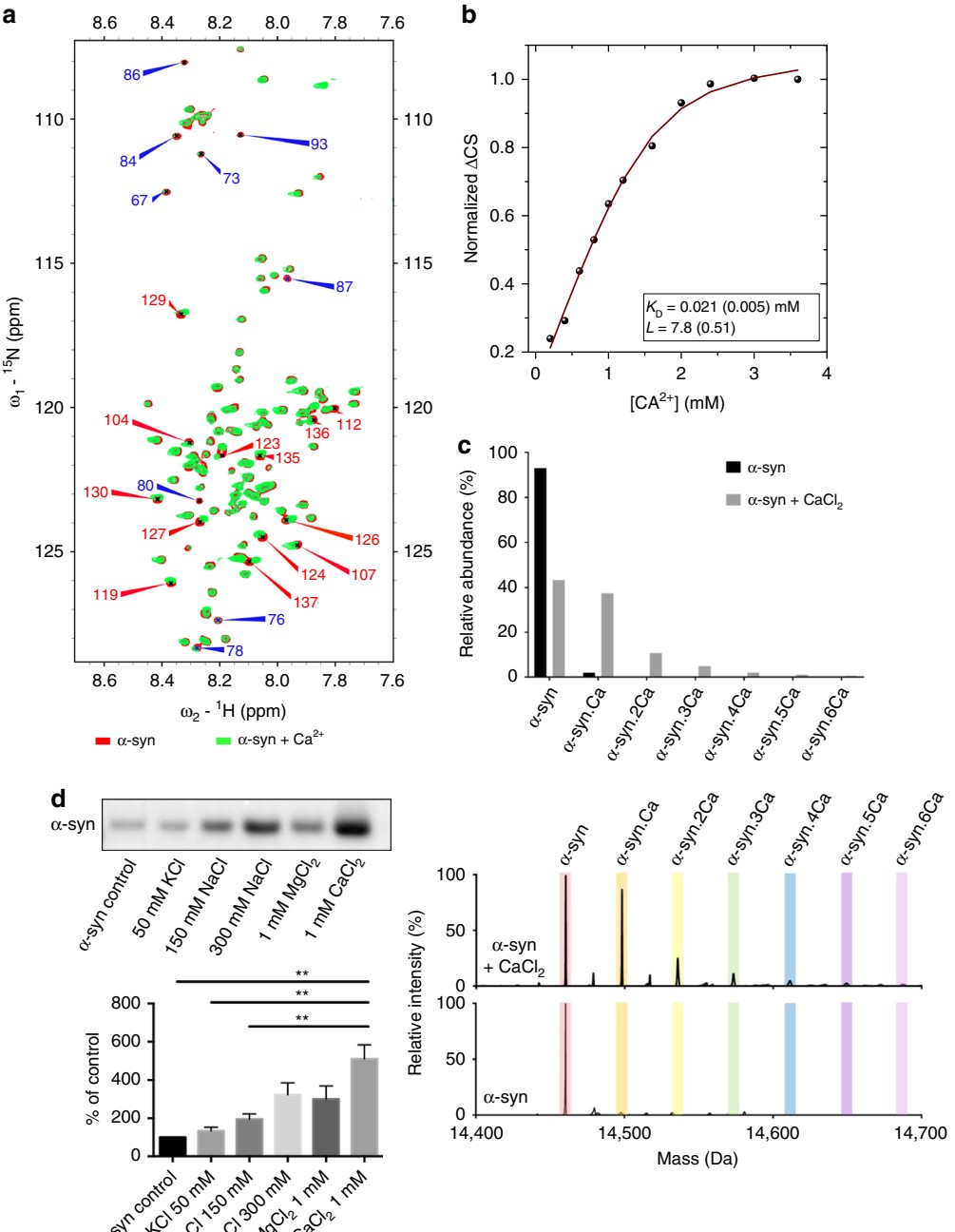

**Fig. 1** Calcium binding to the C terminus of alpha-synuclein and lipid binding. **a** $^{1}$H-$^{15}$N HSQC NMR spectrum of alpha-synuclein in the absence (red) and in the presence of calcium (green, 1.6 mM calcium). Major chemical shift perturbations in the presence of calcium are located at the C terminus of alpha-synuclein (red arrows with assigned amino acid residues), whereas peak broadening (blue arrows with assigned amino acid residues) can be seen within the NAC-region. **b** Fitting of alpha-synuclein calcium binding ($K_D$) from $^{1}$H-$^{15}$N HSQC NMR spectra at increasing calcium concentrations, where $L$ indicates the number of Ca$^{2+}$ ions interacting with one alpha-synuclein molecule. **c** Calcium-bound alpha-synuclein species directly observed by mass spectrometry. Electrospray ionization mass spectra were acquired under identical instrument conditions for samples incubated with or without calcium. Multiple alpha-synuclein species were observed upon charge deconvolution of the ion envelope for the 9$^{+}$–19$^{+}$ charge states, inclusive. The masses correspond to alpha-synuclein: calcium complexes up to a stoichiometry of 1:6. **d** Lipid pull-down experiment using lipids from Folch brain extracts, recombinant alpha-synuclein and various ions. Western blot of the amount of protein pulled down shows that more alpha-synuclein was pulled down by the lipids in the presence of calcium. Neither potassium, sodium, nor magnesium increased alpha-synuclein lipid binding to the same extent. **p = 0.0011, 0.0022, and 0.0090 for comparison of 1 mM CaCl$_2$ with alpha-syn control, 50 mM KCl, and 150 mM NaCl, respectively. Calculated using one-way ANOVA with Tukey's post-hoc correction, graphs indicate mean ± s.e.m. $N = 3$ for all groups, corresponding to three biological repeats, d.f. 12

interaction with synaptic vesicles. To determine whether a calcium-dependent lipid interaction is transient, and therefore dynamically regulated, we again performed lipid pull-down experiments. We show that, consistent with the fast exchange regime observed in solution NMR, the calcium-dependent interaction of alpha-synuclein is dynamic, as addition of the calcium chelator EGTA reverses calcium-mediated alpha-synuclein lipid binding (Fig. 2b).

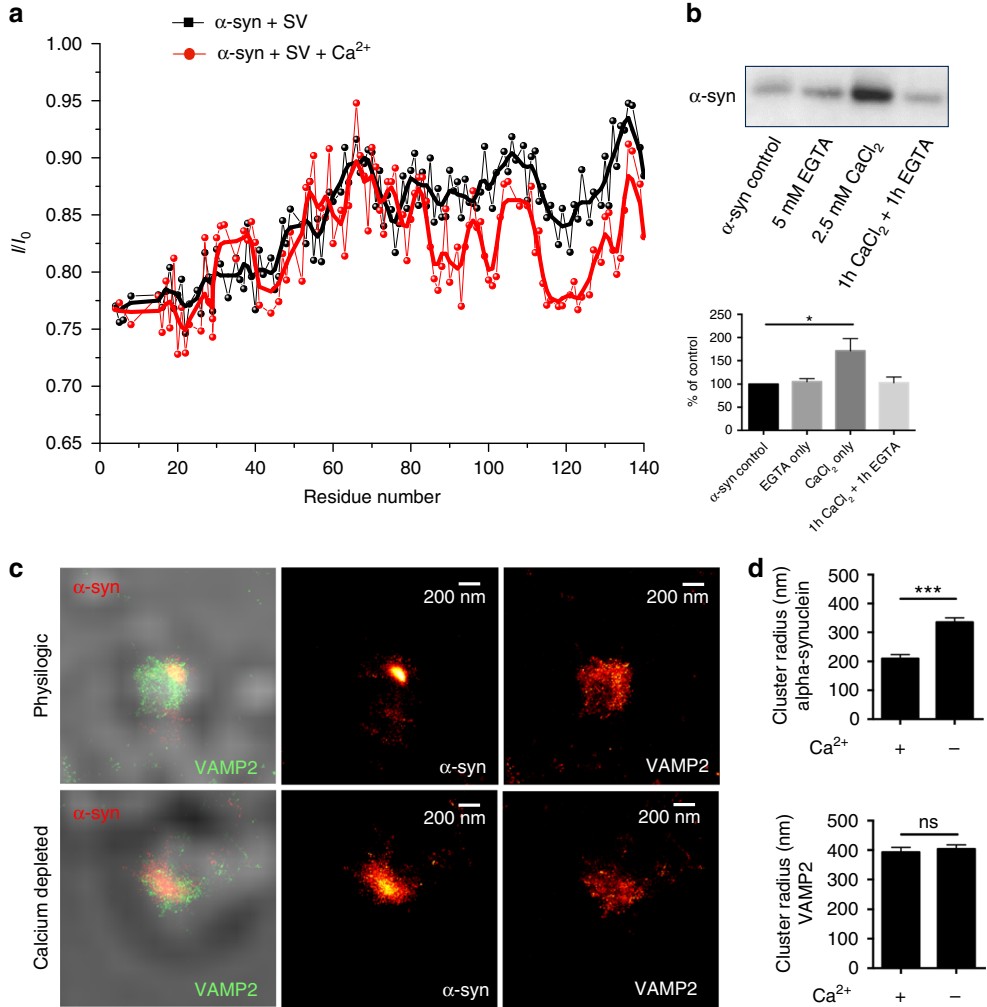

**Fig. 2** The C terminus of alpha-synuclein binds to synaptic vesicles upon calcium binding. **a** CEST-NMR experiments were performed on alpha-synuclein and synaptic vesicles in the absence (black) or presence of calcium (red, 6 mM). In the absence of calcium, the N terminus shows the strongest interaction with synaptic vesicles. Upon addition of calcium, the interaction of the C terminus and also of some residues of the NAC-region increases, which is seen as a reduction of the signal. Experiments were repeated twice. **b** Lipid pull-down experiment showing the transient nature of alpha-synuclein lipid binding. Western blot of the amount of alpha-synuclein pulled down by the lipids showing that calcium-induced lipid binding of alpha-synuclein is reversible upon addition of the calcium chelator EGTA. *$p = 0.0263$, calculated using one-way ANOVA with Tukey's post-hoc correction, graphs indicate mean ± s.e.m. $N = 6$ for control and $CaCl_2$, $n = 4$ for EGTA and $CaCl_2 + $ EGTA, data from three biological repeats, d.f. 16. **c** dSTORM super-resolution imaging of alpha-synuclein and VAMP2 on isolated synaptosomes displaying alpha-synuclein clustering under normal physiological conditions with 2.5 mM calcium in the extracellular buffer (upper panel). Upon calcium depletion in the extracellular buffer, using 1 mM EGTA, alpha-synuclein localization was dispersed (lower panel). **d** Cluster analysis of alpha-synuclein and VAMP2 immunostaining showing increased cluster size of alpha-synuclein upon calcium depletion, whereas VAMP2 cluster size is the same either in the presence of calcium or upon calcium depletion in the extracellular buffer. ****$p < 0.0001$, ns$p = 0.6363$ calculated using two-tailed $t$-test, graphs indicate mean ± s.e.m. $N = 22$ for $+Ca^{2+}$ and $n = 30$ for $-Ca^{2+}$, where $n$ indicates single synaptosomes, data form three biological repeats, d.f. 50

**Alpha-synuclein is modulated by calcium at pre-synaptic terminals**. Synaptosomes, pinched off synapses that reseal as spherical droplets, were isolated from rat brain and used to study the synaptic localization of alpha-synuclein in the presence or absence of calcium. Using direct stochastic optical reconstruction microscopy (dSTORM), permitting sub-diffraction resolution imaging, we found that under normal physiological conditions (low intracellular and high extracellular calcium concentrations) alpha-synuclein was significantly polarized. This could be interpreted as if alpha-synuclein does not bind to synaptic vesicles, however, our vitro data as well as data from previous reports[5–10] show a strong binding of alpha-synuclein to synaptic vesicles. The polarization could therefore be interpreted as a binding of alpha-synuclein to only a subset of synaptic vesicles (similar to what has

been indicated in Lee et al.[19]). When we depleted calcium in the extracellular buffer by the addition of the calcium chelator EGTA and omitting calcium, alpha-synuclein displayed a dispersed distribution throughout the synaptosome (Fig. 2c; Supplementary Fig. 4).

The cluster sizes of the synaptosomal alpha-synuclein and synaptic vesicle associated protein 2 (VAMP2), were determined by cluster analysis and revealed that alpha-synuclein was significantly more dispersed upon calcium starvation with EGTA, while calcium starvation had no effect on the localization of VAMP2 (Fig. 2d). To address whether an increase in intracellular calcium leads to a further clustering of alpha-synuclein, we stimulated the synaptosomes with an extracellular solution at 70 mM KCl, which leads to membrane depolarization and

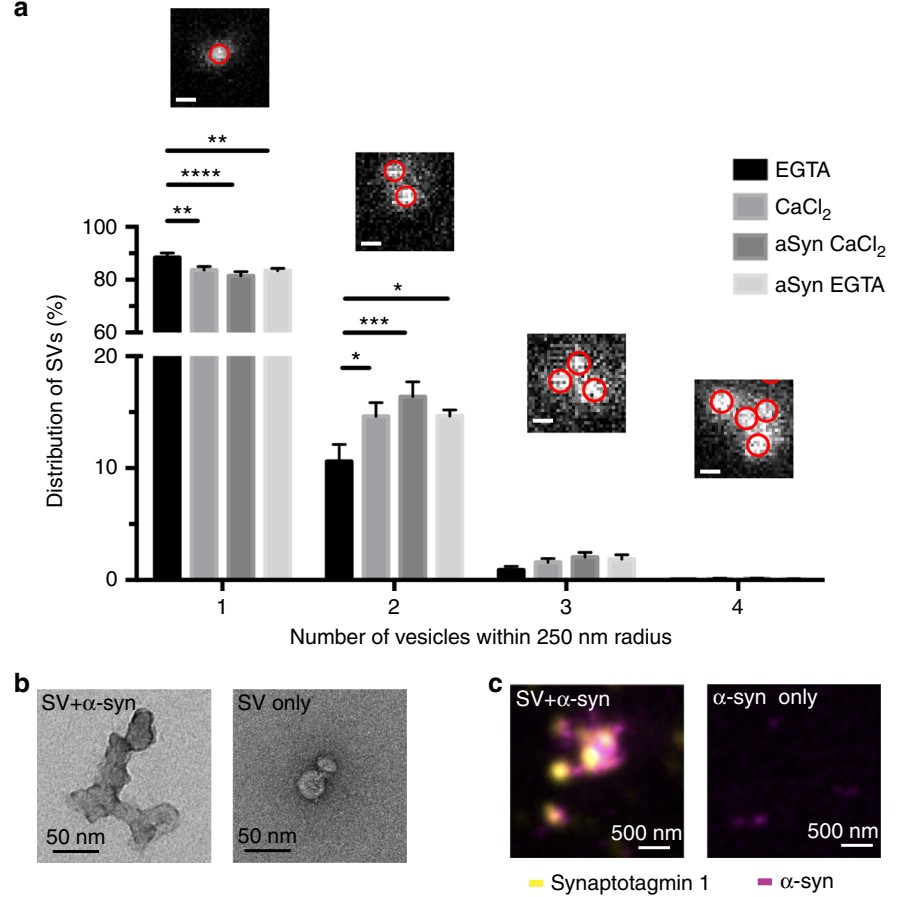

**Fig. 3** Alpha-synuclein and calcium balance the interaction of synaptic vesicles. **a** STED super-resolution imaging of isolated synaptic vesicles incubated with 1 mM EGTA, 200 μM calcium, 50 μM alpha-synuclein + 200 μM calcium, or 50 μM alpha-synuclein + 1 mM EGTA. Images show synaptic vesicles circled in red as detected for analysis of synaptic vesicle clustering. Scales represent 200 nm. Synaptic vesicle clustering is shown as a decrease of synaptic vesicles found as single vesicles and as an increase of synaptic vesicles found in clusters was seen upon incubation of synaptic vesicles with either increased calcium or alpha-synuclein. Note, EGTA was not able to reduce synaptic vesicle clustering in the presence of increased alpha-synuclein concentrations. **p = 0.0023, ****p < 0.0001, and **p = 0.0012 for comparison of % of single vesicles, *p = 0.0140, ***p = 0.0001, and *p = 0.0136 for comparison of % of vesicle clusters of two. Calculated using two-way ANOVA with Tukey's post-hoc correction, graphs indicate mean ± s.e.m. N = 18 for all conditions, data from three biological repeats, d.f. 272. **b** TEM images of synaptic vesicles showing synaptic vesicle clustering in the presence of 50 μM alpha-synuclein. **c** Combined imaging of synaptic vesicles (confocal) and ATTO-647N-labeled alpha-synuclein (STED) showing synaptic vesicles surrounded and glued together by alpha-synuclein. Two biological repeats

calcium influx via voltage-gated calcium channels. Our results show that upon stimulation there is no further change in the distribution of alpha-synuclein throughout the synaptosomes (Supplementary Fig. 5), suggesting that normal physiological calcium concentrations are sufficient to induce alpha-synuclein clustering. The specificity of the alpha-synuclein staining protocol was validated using synaptosomes derived from alpha-synuclein knockout mice (Supplementary Fig. 6). Furthermore, using synaptosomes from wild-type human alpha-synuclein overexpressing mice, we see that alpha-synuclein loses its ability to be specifically polarized under normal physiological conditions (Supplementary Fig. 7).

**Alpha-synuclein and calcium regulate synaptic vesicle interaction.** Using NMR and synthetic lipids, we have shown recently that alpha-synuclein is capable of tethering vesicles and two protein regions were identified as participants in a double anchor mechanism[12]. To quantify the dependencies of synaptic vesicle clustering on calcium and alpha-synuclein, synaptic vesicles purified from rat brains were incubated with either 1 mM EGTA omitting calcium or 200 μM calcium and subsequently imaged using stimulated emission depletion (STED) microscopy. For samples incubated with EGTA 88% of synaptic vesicles were distributed as single vesicles, whereas 12% showed clustering with one or more vesicles. However, in the presence of calcium the amount of single vesicles decreased to 84% and the number of multiple vesicles clustering together increased to 16%. This indicates that, even for synaptic vesicles isolated from wild-type rat brain with endogenous levels of alpha-synuclein present, calcium is a modulating factor of the cohesion between synaptic vesicles. Next, we incubated synaptic vesicles with 200 μM of calcium and 50 μM of recombinant alpha-synuclein, which, in combination with the endogenous alpha-synuclein, leads to doubling of the level of alpha-synuclein present on synaptic vesicles[13]. Again, a significant clustering of synaptic vesicles was observed with 81% distributed as single synaptic vesicles and 19% as multiple vesicle clusters. Incubating synaptic vesicles with 50 μM of recombinant alpha-synuclein and 1 mM EGTA in combination, did however not reverse synaptic vesicle clustering, indicating that the presence of an increased alpha-synuclein level on its own can already induce synaptic vesicle clustering (Fig. 3a).

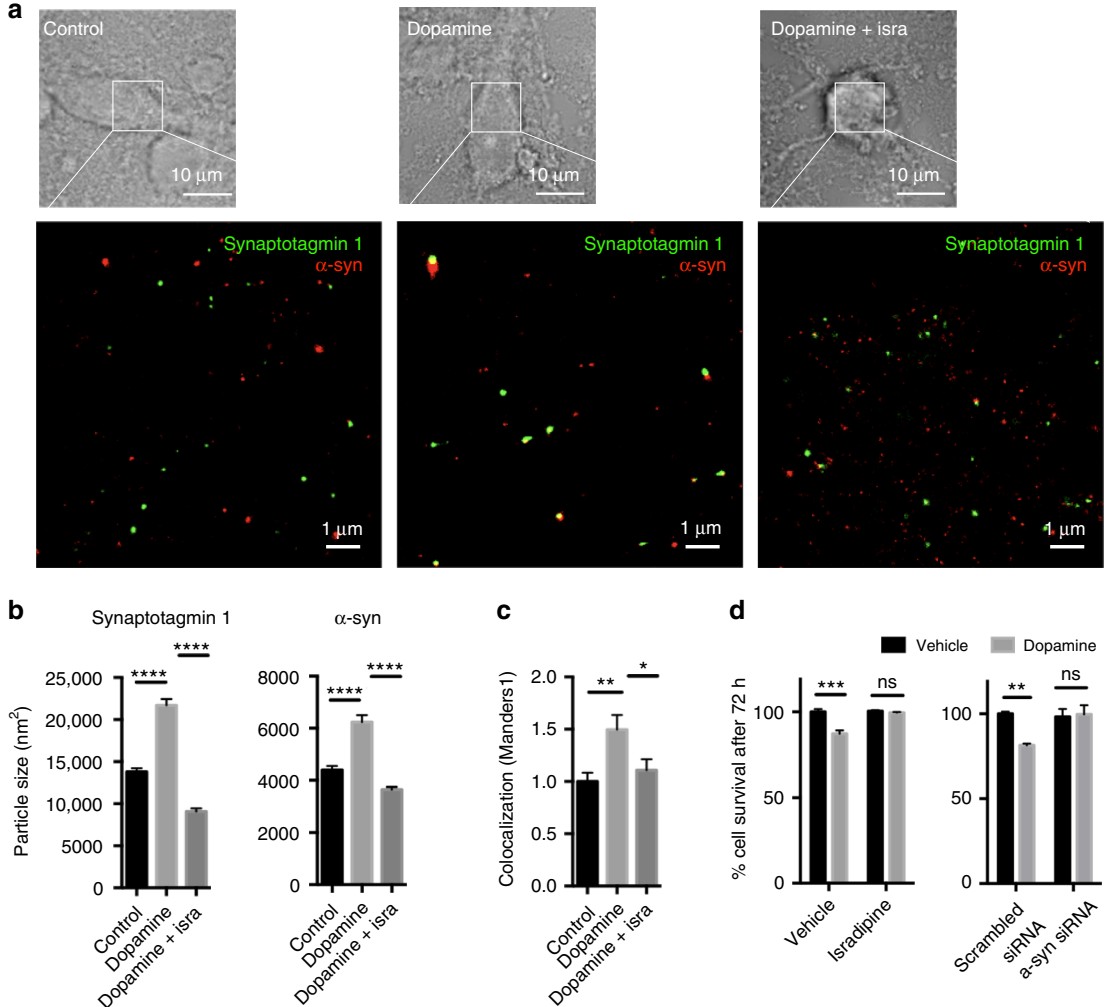

**Fig. 4** Calcium and alpha-synuclein levels mediate dopamine toxicity. **a-c** Ventral midbrain neurons were incubated with 100 µM dopamine in the presence or absence of 5 µM isradipine. *d*STORM super-resolution microscopy of alpha-synuclein and synaptotagmin-1 after 72 h revealed an increase in the area of alpha-synuclein puncta, an increase in the size of synaptotagmin-1 puncta and an increased co-localization of alpha-synuclein with synaptotagmin-1 upon dopamine treatment. These effects were reversed by the Ca$_V$1.3 calcium channel antagonist isradipine, showing decreased size of alpha-synuclein and synaptotagmin-1 puncta and decreased co-localization. ****$p < 0.0001$ for synaptotagmin and alpha-synuclein, **$p < 0.0066$, *$p < 0.0446$ for co-localization, calculated using one-way ANOVA with Tukey's post-hoc correction, graphs indicate mean ± s.e.m. $N = 2251, 1154, 1987$ for synaptotagmin, d.f. 5353, $n = 5198, 3210, 6845$ for alpha-synuclein, d.f. 15250, where $n$ indicates individual clusters identified from 30, 30, 29 images from three biological repeats, $n = 30, 30, 29$ for co-localization, where $n$ indicates number of images. **d** Dopamine toxicity in SH-SY5Y cells after 72 h incubation with 100 µM dopamine was rescued upon treatment with 5 µM isradipine and upon alpha-synuclein knockdown, showing that both, calcium and alpha-synuclein are necessary for toxicity to occur. ***$p < 0.0007$, $^{ns}p = 0.9935$ for isradipine, $n = 12$ for all groups, where $n$ indicates number of wells, d.f. 44; **$p < 0.0062$, $^{ns}p = 0.9934$ for alpha-synuclein knockout, $n = 8$ for all groups, where $n$ indicates wells, d.f. 28, calculated using one-way ANOVA with Tukey's post-hoc correction, graphs indicate mean ± s.e.m. Three biological repeats

A clustering of synaptic vesicles upon incubation with 50 µM of alpha-synuclein was also observed via transmission electron microscopy (Fig. 3b) and via combined confocal/STED imaging, the latter verifying that synaptic vesicles colocalized with alpha-synuclein (Fig. 3c).

Next, we studied the behavior of endogenous alpha-synuclein in ventral midbrain (VM) neurons incubated with 100 µM dopamine. This system has previously been shown to induce the formation of alpha-synuclein oligomers and to exhibit dopaminergic neuron-specific toxicity[20–23]. VM neurons treated with dopamine for 72 h were stained against endogenous alpha-synuclein and the synaptic vesicle protein synaptotagmin-1 and subsequently imaged by *d*STORM. Upon incubation with dopamine, an increase in the area of alpha-synuclein-positive puncta was observed, together with an increased size of

synaptotagmin-1 puncta and an increased co-localization of alpha-synuclein and synaptotagmin-1 (Fig. 4a–c). We show that these dopamine-induced changes were reversed by treating the cells with isradipine, a Ca$_V$1.3 calcium channel antagonist previously reported to block dopaminergic neuron cell death in 1-methyl-4-phenyl-1,2,3,6-tetrahydropyridine (MPTP) and other neurotoxin-related Parkinson's disease models[24]. This demonstrates that in the presence of endogenous alpha-synuclein levels calcium is necessary to induce synaptic vesicle clustering, consistent with our observations from isolated synaptic vesicles. As isradipine was seen to significantly reduce clustering of alpha-synuclein, we tested if also dopamine-induced cytotoxicity might be suppressed as a consequence of its action. Our data show that dopamine-induced toxicity can be abolished both by the administration of isradipine, or by knocking down of alpha-

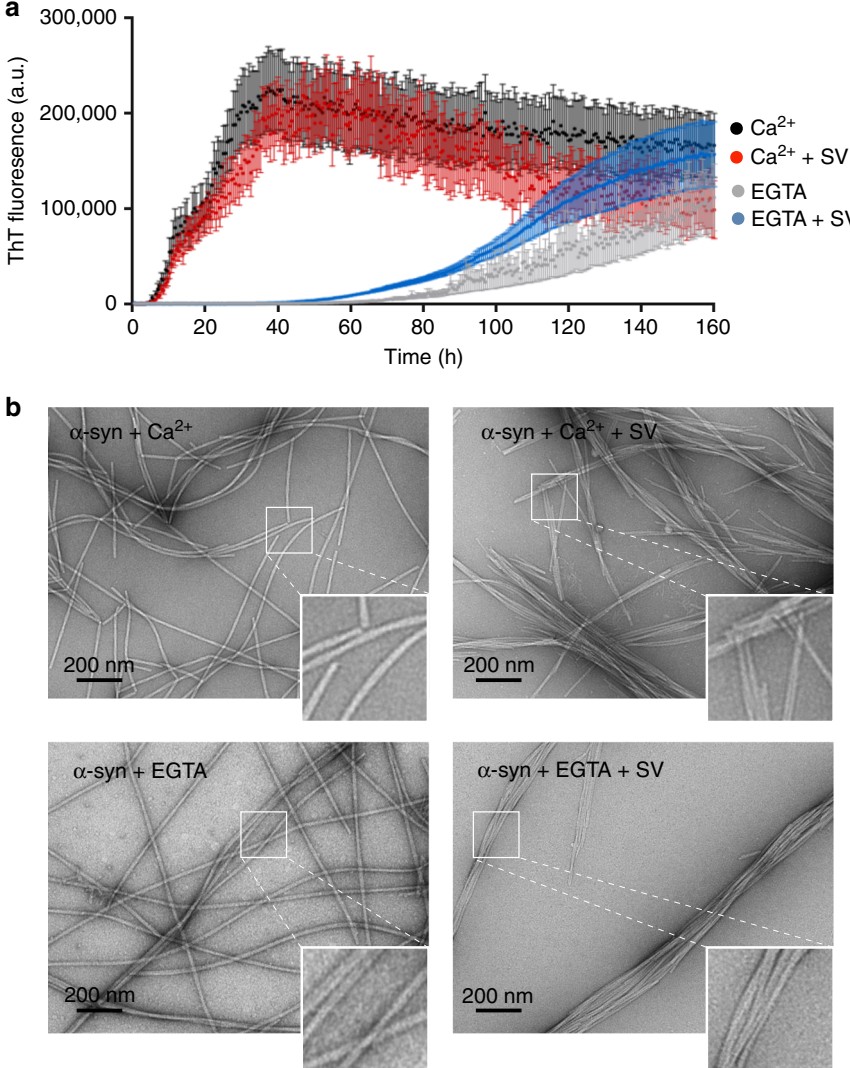

**Fig. 5** The effect of calcium and synaptic vesicles on alpha-synuclein aggregation. **a** Alpha-synuclein aggregation measured by ThT fluorescence using 100 μM monomeric alpha-synuclein under shaking conditions. The presence of 2.5 mM calcium increased the aggregation kinetics of alpha-synuclein compared to 1 mM EGTA, both in the presence or absence of synaptic vesicles (black and red). For the EGTA-containing groups, there was a trend towards faster aggregation of alpha-synuclein in the presence of synaptic vesicles (blue vs. gray). Three biological repeats, $n = 18$ for all conditions, where $n$ represents single wells. values represent mean ± s.e.m. **b** TEM images of alpha-synuclein fibrils formed in the presence of 2.5 mM calcium or 1 mM EGTA either in the presence or absence of synaptic vesicles. Differences in the morphology of alpha-synuclein fibrils were observed for fibrils formed in the presence of 2.5 mM calcium in the absence and presence of synaptic vesicles. In the presence of synaptic vesicles, alpha-synuclein fibrils showed increased lateral bundling and shortening. Alpha-synuclein aggregation in the presence of 1 mM EGTA led to substantially less fibrils, however the fibrils formed, retained their morphological phenotype compared to the fibrils formed in the presence of 2.5 mM calcium. Alpha-synuclein fibrils formed in the presence of synaptic vesicles plus 1 mM EGTA showed an intermediate phenotype, with bundled, but more elongated fibril structures than found in the synaptic vesicles and 2.5 mM calcium group. Experiments were repeated twice

synuclein (Fig. 4d; Supplementary Fig. 8), indicating that high levels of calcium and/or alpha-synuclein are key elements of neuronal toxicity.

**Calcium and synaptic vesicles affect alpha-synuclein aggregation.** Taken together our data indicate that calcium mediates the interaction of alpha-synuclein and synaptic vesicles and that this has an effect in both physiological processes and under conditions generating cellular toxicity. However, since both, either increased levels of alpha-synuclein or calcium can induce cell death, we investigated whether calcium and synaptic vesicles may influence the aggregation propensity of alpha-synuclein. To test this hypothesis, we performed Thioflavin T (ThT) fluorescence assays

of alpha-synuclein aggregation, and analyzed the effect of increased calcium in the presence and absence of synaptic vesicles (Fig. 5a). The lag time for each condition was calculated from the averaged aggregation curves by linear extension of the elongation phase. The nucleation rate ($k_1$) and the elongation rate ($k_2$) were fitted using the Finke–Watzky equation for a two-step aggregation mechanism[25]. This analysis clearly revealed that calcium aggravates alpha-synuclein aggregation, showing a lag time decrease from 79 h for "EGTA only" to 4 h for "calcium only". Accordingly, the nucleation rate was increased 2-fold in the presence of calcium, and the elongation rate was increased 1.3-fold. Moreover, the amount of residual monomer left at the end of the aggregation assay was significantly lower when calcium was present, confirming the higher aggregation propensity of

**Table 1 The effect of calcium and synaptic vesicles on alpha-synuclein aggregation**

| Condition | Lag time (h) | $k_1$ (m/s) | $k_2$ (per s/%int) | Remaining monomer (μM) |
|---|---|---|---|---|
| Ca²⁺ only | 3.95±0.53 | 0.24±0.69 | 8.07±1.28 | 8.5±0.3**** |
| Ca²⁺ + SV | 5.55±2.59 | 0.78±5.33 | 15.92±5.02 | —ᵃ |
| EGTA only | 78.95±0.42 | 0.09±0.04 | 6.32±0.65 | 67.8±5.4 |
| EGTA + SV | 44.72±0.42 | 0.22±0.06 | 5.50±0.51 | —ᵃ |

ThT fluorescence intensity profiles were used to calculate the lag time as well as the nucleation ($k_1$) and the elongation rate ($k_2$). Lag time is given as X intercept with standard error. $k_1$ and $k_2$ are given as constants with a 95% confidence interval. Three biological repeats. Remaining alpha-synuclein monomer concentration was revealed by analytical SEC-HPLC at the end of the experiment
****$p < 0.0001$, $n = 10$ for Ca²⁺ only, and $n = 9$ for EGTA only, where $n$ indicates wells, two biological repeats, calculated using two-tailed $t$-test, values indicate mean ± s.e.m. d.f. 17
ᵃRemaining monomer concentration could not be calculated due to presence of protein from synaptic vesicles preventing clear detection of remaining asyn monomer

alpha-synuclein in the presence of calcium. In the presence synaptic vesicles and calcium, we observed the highest nucleation and elongation rate. In the absence of calcium, synaptic vesicles decreased the lag time from 79 to 44 h and increased the nucleation rate 2-fold compared to the "EGTA only" group, which is in accordance with what has been reported for synthetic vesicles[17]. However, the elongation rate was not increased (Table 1).

Interestingly, alpha-synuclein fibrils formed at the end of the assay showed a different morphology when synaptic vesicles were present during the aggregation process. When alpha-synuclein was incubated in the presence of calcium and synaptic vesicles, alpha-synuclein fibrils were shorter and showed aggravated bundling. In the presence of EGTA significantly less fibrils were found. The few fibrils that could be found in the EGTA group without synaptic vesicles were long and separated whereas the EGTA fibrils formed in the presence of synaptic vesicles appeared to have an intermediate phenotype as they were still bundled but longer than the fibrils formed in the presence of synaptic vesicles and calcium (Fig. 5b). To confirm the presence of synaptic vesicles during the aggregation process we show a TEM image of alpha-synuclein in the presence of synaptic vesicles at the beginning of the aggregation experiment (Supplementary Fig. 9), as after seven days the vesicles cannot be detected on the TEM anymore.

## Discussion
The C terminus of alpha-synuclein is negatively charged such that electrostatic interaction can take place with cations[16]. In the past, however, research on such interactions has predominantly been focused on metal ions as environmental factors inducing alpha-synuclein aggregation[26–28]. Here, we focus on the interaction of alpha-synuclein with physiological calcium, since at the pre-synaptic terminal, where alpha-synucleins primarily resides, large fluctuations in calcium levels are known to occur[14,15]. We quantify and localize the interactions of calcium with alpha-synuclein using ¹H-¹⁵N HSQC NMR and observe significant chemical shift perturbations for a number of residues at the C terminus of the protein. Furthermore, significant peak broadening takes place in the NAC-region of alpha-synucein in the presence of calcium. This can be interpreted either as a conformational change in the NAC-region induced upon calcium binding at the C terminus, or as an interaction between multiple NAC-regions when individual alpha-synuclein molecules cluster due to charge neutralization. The affinity of calcium was found to be around 21 μM, which is lower than reported for other multi-valent cations[27]. Although levels of calcium are in the tens of nM range under resting conditions[29], it's concentration rises to several hundred μM within microdomains during depolarization of neurons as a result of a concomitant calcium influx via voltage-gated calcium channels[14,15]. In this context, the observed $K_D$ of

21 μM highlights the physiological relevance of the interaction between alpha-synuclein and calcium.

CEST-NMR experiments, analyzing the interaction of alpha-synuclein with synaptic vesicles from rat brain, verified that in the presence of calcium the C terminus has a higher affinity to bind to synaptic vesicles. In a previous study[12] we have shown that the modes of binding of the N-terminal and NAC-region of alpha-synuclein to small unilammelar vesicles (SUVs) of DOPE:DOPS:DOPC (in 5:3:2 molar ratios)[11,12,30] are independent of each other. This degree of independence suggested that the two regions can bind the same vesicle but also multiple vesicles. We referred to this N terminus and NAC-region binding of alpha-synuclein to SUVs as double anchor mechanism. We now have characterized the binding of alpha-synuclein to synaptic vesicles isolated from rat brain using CEST-NMR, showing again that the strongest binding occurs at the N terminus and revealing an intermediate level of synaptic vesicle interaction of the NAC-region and the C terminus. Interestingly, upon addition of calcium this binding via the C terminus, and also for certain residues of the NAC-region, was found to be increased. We refer to this as extended double anchor mechanism since the double anchor has been extended to the C terminus of alpha-synuclein. Our NMR data are consistent with a study using site specific pyrene labeling of alpha-synuclein[31]. In this study, it was shown that the N terminus of alpha-synuclein binds to synthetic vesicles in the absence of calcium, while the C terminus does not. Upon calcium addition on its own, there is a reduction in the polarity of the C terminus, which suggests calcium binding. Moreover, upon addition of synthetic vesicles a further reduction in the polarity was observed, supporting a calcium-dependent lipid binding of the C terminus. Summarizing the above, alpha-synuclein, in the presence of calcium, may act as an extended and strengthened double anchor between synaptic vesicles, which could cause interaction in three different ways: (i) the N terminus and C terminus both tether to the same vesicle, (ii) the N terminus binds to one vesicle, whereas the C terminus binds to another vesicle via an extended double anchor mechanism, or (iii) the N terminus binds to synaptic vesicles, whereas the C terminus binds to the plasma membrane. Our observations, using synaptosomes, isolated synaptic vesicles as well as ventral mesencephalic cells, suggest that calcium and alpha-synuclein can affect vesicle pool homeostasis, either via promoting intervesicular interactions and/or via tethering of synaptic vesicles to the plasma membrane, which could influence their proximity to voltage-gated calcium channels[32].

From the time of its discovery, alpha-synuclein has been known as a pre-synaptic protein, suggesting a role in neurotransmitter release. It has been shown that the overexpression of alpha-synuclein inhibits neurotransmitter release[33–37], whereas neurotransmitter release appears facilitated in an alpha-synuclein knockout model[38–41]. However, data are not conclusive so far and a regulatory role rather than a direct role of alpha-synuclein in exocytosis is still debated in the literature[42]. It has furthermore

been suggested that alpha-synuclein has a role in endocytosis and in synaptic vesicle homeostasis (for detailed review see ref. [43]).

We show here, that in synaptosomes alpha-synuclein localization is dependent on calcium since calcium depletion in the extracellular space does not lead to the polarization of alpha-synuclein in the synaptosomes. It is interesting to note that we see little overlap between alpha-synuclein and VAMP2, which could be interpreted as if alpha-synuclein does not bind to synaptic vesicles at all. We do, however, not believe that this is the case for two main reasons: First, all our in vitro data as well as data from others[5–10] show strong binding of alpha-synuclein to synaptic vesicles. Second, it has been suggested that alpha-synuclein binds to only a subset of synaptic vesicles[19], however, it may require further investigations to fully determine which synaptic vesicles are positive for which synaptic vesicle protein.

We show that calcium as well as increased alpha-synuclein concentrations influence the clustering of synaptic vesicles in vitro. Similar observations were made using synaptic vesicle-mimics built of anionic phospholipids containing SNARE complex proteins[44]. The authors showed that an increased alpha-synuclein concentration caused clustering of the synaptic vesicle-mimics, which was dependent on the capacity of alpha-synuclein to bind to lipids, as the lipid-binding deficient familial A30P alpha-synuclein mutant displayed a decreased propensity to cluster vesicles. The authors report that also C-terminally truncated alpha-synuclein can decrease vesicle clustering, which supports our hypothesis of an extended double anchor binding mechanism of alpha-synuclein. The authors though indicate that this is mediated via C-terminal binding of alpha-synuclein to VAMP2. In our study, however, we do not observe clustering of VAMP2, suggesting that a VAMP2-independent mechanism might be involved. Our proposed model of alpha-synuclein-dependent vesicle clustering is further supported by recent studies demonstrating a decrease in synaptic vesicle motility in neurons upon alpha-synuclein overexpression[45,46]. This is in line with our observation that the overexpression of wild-type human alpha-synuclein reduced the propensity of alpha-synuclein to be polarized upon calcium treatment in synaptosomes. It is important to note though that not all of these calcium-controlled processes rely solely on the presence of alpha-synuclein. This becomes clear from alpha-synuclein knockout studies in mice[41,47] but is further supported by the fact, that alpha-synuclein is expressed at late stages of development, as shown for songbirds, for which alpha-synuclein becomes upregulated during song acquisition[48].

Furthermore, we show here that dopamine toxicity known to induce alpha-synuclein oligomerization[20–23] can be significantly reduced if either calcium levels are decreased, using the Ca$_V$1.3 blocker isradipine, or if alpha-synuclein levels are reduced by knocking down alpha-synuclein, suggesting that both calcium and alpha-synuclein need to be present to convey dopamine-induced toxicity. This is in line with the findings that dopaminergic neurons of the substantia nigra (SN) display a lower abundance of calcium-binding proteins[49] and exhibit a calcium-driven pacemaking activity[50], putting them at higher risk for calcium-mediated pathophysiology. Using in vitro aggregation assays, we see that calcium clearly aggravates alpha-synuclein aggregation, in the presence or absence of synaptic vesicles. We also observed an increase in nucleation when synaptic vesicles were added to alpha-synuclein with EGTA, which is in line with the literature reporting that synthetic lipids increase the rate of aggregation of alpha-synuclein[17]. Excess calcium can cause a conformational change within the protein, such as by an exposure of the NAC-region upon calcium binding, or simply via a change in the net charge of the protein, both facilitating the propensity for aggregation. It is interesting to note though that alpha-synuclein fibrils formed in vitro in the presence of isolated synaptic vesicles display a different morphology.

Understanding the above described role of alpha-synuclein in physiological or pathological processes may impact strongly on the development of new therapeutics for PD. Interestingly, antibodies targeted to the C terminus of alpha-synuclein were demonstrated to decrease intracellular alpha-synuclein pathology in animal models of synucleinopathy[51]. One such antibody, PRX002, is currently in phase 2 clinical development for Parkinson's disease (ClinicalTrials.gov Identifier: NCT03100149). Also isradipine, which is used as calcium channel blocker in heart diseases, may prove to be a valuable candidate to act against PD via lowering intracellular calcium load[52,53] (ClinicalTrials.gov Identifier: NCT02168842).

## Methods

**Purification of alpha-synuclein.** Human wild-type (WT) alpha-synuclein was expressed in Escherichia coli One Shot® BL21 STAR™ (DE3) (Invitrogen, Thermo Fisher Scientific, Cheshire, UK) cells using plasmid pT7-7 and purified using ion exchange on a HiPrep Q FF 16/10 anion exchange column (GE Healthcare, Uppsala, Sweden)[54]. Alpha-synuclein was then further purified on a HiPrep Phenyl FF 16/10 (High Sub) hydrophobic interaction column (GE Healthcare)[55]. Purification was performed on an ÄKTA Pure (GE Healthcare, Sweden). Monomeric protein was dialyzed against 20 mM Na$_2$HPO$_4$ pH 7.2 and stored at −80 °C. For experiments with dye-labeled alpha-synuclein, the cysteine mutant N122C was purified and labeled with the ATTO-647N dye (#05316, Sigma-Aldrich, Dorset, UK). For experiments with vesicles, monomeric human WT alpha-synuclein was buffer exchanged using PD10 Desalting Columns (GE Healthcare) into vesicle buffer (20 mM NaCl, 2.5 mM KCl, 25 mM HEPES, 30 mM Glucose, pH 7.4 with NaOH).

**Lipid pull-down assay.** Lipid extract from bovine brain (Type I, Folch Fraction I; Sigma-Aldrich) was dissolved in lipid buffer (1 mg/mL; 50 mM Tris + 100 mM NaCl, pH 7.4) and sonicated on ice using a Branson SLPe sonicator (Branson Ultrasonic S.A., Geneva, Switzerland). The lipid was incubated with 2 µg of recombinant human alpha-synuclein with either KCl (50 mM), NaCl (150 or 300 mM), MgCl$_2$ (1 mM) or CaCl$_2$ (1 mM), for 1 h at room temperature (RT). The different samples were centrifuged for 20 min at 4000 ×g at RT to obtain the pellet for analysis by Western blot. The lipid pull-down assay to prove reversibility of Ca$^{2+}$-dependent lipid binding was done using post-incubation with EGTA (5 mM) for one 1 h.

Western blot of alpha-synuclein was performed using 4–12% Bis-Tris gels (Life Technologies), the protein was transferred onto 0.2 µm Millipore PVDF membrane (Fisher Scientific, Loughborough, UK) and subsequently fixed using 4% formaldehyde (Sigma-Aldrich) and 0.1% glutaraldehyde (Sigma-Aldrich). The primary mouse anti-alpha-synuclein antibody LB509 (LB509, 1:1000 dilution, Life Technologies) and an enhanced chemoluminescence (ECL)-horse radish peroxidase (HRP) conjugated secondary antibody (NA931, 1:1000 dilution, GE Healthcare) and SuperSignal West Femto Chemiluminescent Substrate (Thermo Fisher Scientific, Epsom, UK) were used to probe the membrane, which was exposed using a G:BOX (Syngene, Cambridge, UK).

**Animals.** Animals were bred and supplied by Charles River UK Ltd., Scientific, Breeding and Supplying Establishment, registered under Animals (Scientific Procedures) Act 1986, and AAALAC International accredited. All animal work conformed to guidelines of animal husbandry as provided by the UK Home Office. Animals were sacrificed under schedule 1; procedures that do not require specific Home Office approval. Animal work was approved by the NACWO and University of Cambridge Ethics Board.

**Synaptic vesicles, synaptosomes, and cell cultures.** Isolation of synaptic vesicles (SV) was performed as previously described[56]. Brains were dissected from WT Sprague-Dawley rats, two brains were used per SV preparation. The vesicle pellet was resuspended in 200 µL of vesicle buffer (20 mM NaCl, 2.5 mM KCl, 25 mM HEPES, 30 mM Glucose, pH 7.4 with NaOH) and snap-frozen in aliquots in liquid nitrogen before being stored at −80 °C. SVs had a protein concentration of 3.3 mg/mL.

Synaptosomes were prepared from WT Sprague-Dawley adult rat brains as described previously[57]. Briefly, the brain was homogenized in a glass-Teflon EUROSTAR20 homogenizer (IKA, Oxon, UK) in homogenizing buffer made from sucrose/EDTA buffer (320 mM sucrose, 1 mM EDTA, 5 mM Tris, pH 7.4) with 50 mM DTT, using 10 strokes at 800 rpm. Synaptosomes were isolated using 3–23% Percoll gradients. The synaptosome containing fractions were pooled and resuspended in extracellular buffer solution (20 mM sodium HEPES, 130 mM NaCl, 5 mM NaHCO$_3$, 1.2 mM Na$_2$HPO$_4$, 1 mM MgCl$_2$, 10 mM glucose, 5 mM KCl, 2.5 mM CaCl$_2$, pH 7.4). Synaptosomes were loaded onto eight well glass bottom µ-slides (ibidi GmbH, Munich, Germany), which were cleaned with 1 M

KOH and coated with poly-L-lysine 0.01% solution (mol wt 70,000–150,000, Sigma-Aldrich) over night at 4 °C. Synaptosomes were stimulated for 30 min at 37 °C with extracellular solution made at either 5 mM KCl + 2.5 mM CaCl$_2$, 5 mM KCl + 1 mM EGTA or 70 mM KCl + 2.5 mM CaCl$_2$. Synaptosomes were then fixed with 4% formaldehyde (Sigma-Aldrich) in PBS. Fixation was quenched by washing with 0.1 M glycine in PBS for 5 min.

Ventral mesencephalic (VM) neurons were dissected from E14 Sprague-Dawley rat embryos. In brief, VM tissue was incubated in 0.1% trypsin (Worthington Biochemical Corporation, Lakewood, USA) and 0.05% DNase (Sigma-Aldrich) in DMEM (Sigma-Aldrich) for 20 min at 37 °C. Cells were washed four times with 0.05% DNase in DMEM and triturated until a single cell suspension was reached. Neurons were seeded at 100,000 cells/well in LabTek II chambered coverglass (Thermo Fisher Scientific) coated with poly-L-lysine 0.01% solution (mol wt 70,000–150,000, Sigma-Aldrich). Neurons were kept in DMEM with 10% fetal bovine serum (FBS, 10270-106, Gibco®) for 3 h, then media was changed to Neurobasal media Gibco® supplemented with 2% B27 Gibco®, 0.5 mM GlutaMax Gibco® and 1% antibiotic-antimycotic Gibco® (all Thermo Fisher Scientific). Neurons were used at days in vitro (DIV) 14. Fixation was performed for 10 min using 4% formaldehyde (Sigma-Aldrich) in PBS containing 4% sucrose, 5 mM MgCl$_2$, and 10 mM EGTA.

Human neuroblastoma cells (SH-SY5Y) were obtained from the European Collection of Cell Cultures (ECACC, Sigma-Aldrich) and grown in a 1:1 minimal essential medium (MEM) (Sigma-Aldrich) and nutrient mixture F-12 Ham (Sigma-Aldrich) supplemented with 15% FBS Gibco®, 1% non-essential amino-acids Gibco®, 2 mM GlutaMAX Gibco®, and 1% antibiotic-antimycotic Gibco® (all Thermo Fisher Scientific). Cells were plated at 5000 cells/well in Nunc MicroWell 96-well plates (Thermo Fisher Scientific) for cytotoxicity assays and at 700,000 cells/dish in 48 mm dishes (Nunc A/A) for western blotting studies. Cells were tested for mycoplasma contamination.

Treatment of cells was performed using 100 µM dopamine (100× stock solution in water, freshly prepared, Sigma-Aldrich) or 100 µM dopamine + 5 µM isradipine (1000× stock solution in DMSO, Sigma-Aldrich). Control cells received 0.1% DMSO and 1% water, respectively. Isradipine treatment was carried out 30 min early to dopamine treatment. After 72 h of incubation, cells were fixed or underwent cytotoxicity assay using the cell cytotoxicity assay kit, ab112118 (Abcam, Cambridge, UK) according to manufacturer's instructions. Absorbance intensity was measured at 570 nm and 605 nm, with the ratio OD570/OD605 being proportional to the number of viable cells.

**Solution NMR.** To probe the structure and thermodynamics of calcium binding with alpha-synuclein at a residue specific level, we employed a series of $^1$H-$^{15}$N HSQC experiments using different concentrations of Ca$^{2+}$ (0–3.6 mM) and a fixed concentration of alpha-synuclein (200 µM). NMR experiments were carried out at 10 °C on a Bruker spectrometer operating at $^1$H frequencies of 800 MHz equipped with triple resonance HCN cryo-probe. The $^1$H-$^{15}$N HSQC experiments were recorded using a data matrix consisting of 2048 ($t_2$, $^1$H) × 220 ($t_1$, $^{15}$N) complex points. Assignments of the resonances in $^1$H-$^{15}$N-HSQC spectra of alpha-synuclein were derived from our previous studies[11].

The chemical shift perturbation in the $^1$H-$^{15}$N HSQC spectra was analyzed using a weighting function:

$$\Delta\delta = \sqrt{\frac{1}{2}\left(\delta_{\mathrm{H}}^2 + 0.15\delta_{\mathrm{N}}^2\right)}.$$

These provide the fraction of bound alpha-synuclein, $\chi_{\mathrm{B}}$, which is calculated as:

$$\chi_{\mathrm{B}} = \frac{\Delta\delta_{\mathrm{obs}}}{\Delta\delta_{\mathrm{sat}}}.$$

Where the $\Delta\delta_{\mathrm{obs}}$ is the variation of the chemical shifts of a peak of alpha-synuclein that is observed at a given [Ca$^{2+}$], and $\Delta\delta_{\mathrm{sat}}$ is the maximum variation obtained at saturation with an excess of calcium. $\chi_{\mathrm{B}}$ was calculated as a function of [Ca$^{2+}$] for every peak of the protein, and a global $\chi_{\mathrm{B}}$ was used to include the chemical shift variations from all the peaks associated with the major perturbations in the presence of calcium. We then used a fitting procedure based on a binding model describing $\chi_{\mathrm{B}}$ as a function of the total [Ca$^{2+}$][17]

$$\alpha\mathrm{syn}^{\mathrm{U}} + \mathrm{Ca}_L^{2+} \leftrightarrows \alpha\mathrm{syn}^{\mathrm{B}}\left(\mathrm{Ca}^{2+}\right)_L. \quad (1)$$

Where $\alpha\mathrm{syn}^{\mathrm{U}}$ and $\alpha\mathrm{syn}^{\mathrm{B}}$ indicate free and calcium-bound alpha-synuclein, $L$ indicates the number of Ca$^{2+}$ interacting with one alpha-synuclein molecule. The equilibrium dissociation constant from this model is given by

$$K_{\mathrm{D}} = \frac{\left[\alpha\mathrm{syn}^{\mathrm{U}}\right]\left[\mathrm{Ca}_L^{2+}\right]}{\left[\alpha\mathrm{syn}^{\mathrm{B}}(\mathrm{Ca}^{2+})_L\right]}, \quad (2)$$

the overall concentration of alpha-synuclein in this equilibrium is given by

$$[\alpha\mathrm{syn}] = \left[\alpha\mathrm{syn}^{\mathrm{U}}\right] + \left[\alpha\mathrm{syn}^{\mathrm{B}}\left(\mathrm{Ca}^{2+}\right)_L\right], \quad (3)$$

and the overall concentration of Ca$^{2+}$ is given by

$$[\mathrm{Ca}^{2+}] = L\left(\left[\mathrm{Ca}^{2+}\right]_L + \left[\alpha\mathrm{syn}^{\mathrm{B}}\left(\mathrm{Ca}^{2+}\right)_L\right]\right). \quad (4)$$

Leading to the following fitting function

$$\chi_{\mathrm{B}} = \frac{[\alpha\mathrm{syn}] + \left[\frac{\mathrm{Ca}^{2+}}{L}\right] + K_{\mathrm{D}} - \sqrt{\left([\alpha\mathrm{syn}] + \left[\frac{\mathrm{Ca}^{2+}}{L}\right] + K_{\mathrm{D}}\right)^2 - \frac{4[\alpha\mathrm{syn}][\mathrm{Ca}^{2+}]}{L}}}{2[\alpha\mathrm{syn}]}.$$

**Chemical exchange saturation transfer (CEST) NMR.** CEST-NMR is widely used to probe the interaction of amyloidogenic proteins[58,59]. Here we employed CEST measurements[11,60–63] to directly probe the equilibrium between vesicle unbound and bound states of alpha-synuclein. CEST has enhanced characteristics compared to standard heteronuclear correlation spectroscopy in probing the details of the equilibrium between NMR visible (unbound alpha-synuclein) and NMR invisible (vesicle bound alpha-synuclein). These include a significant sensitivity at low vesicle:protein ratios and the avoidance of other factors that may influence the transverse relaxation rates of the protein resonances. CEST experiments were carried out at 10 °C on a Bruker spectrometer operating at $^1$H frequencies of 800 MHz equipped with triple resonance HCN cryo-probe. The measurements were based on $^1$H-$^{15}$N HSQC experiments by applying constant wave saturation of 400 Hz in the $^{15}$N channel. A series of large offsets was employed (−9, −7, −5, −4, −3, −1.5, 0, 1.5, 3, 4, 5, 7, 9 kHz), and additional spectrum, saturated at −100 kHz, was recorded as a reference. The saturation of the bound state is transferred to the free state via the conformational exchange of these two states, resulting in the saturation of the peak intensities in the visible unbound state. The CEST experiments were recorded using a data matrix consisting of 2048 ($t_2$, $^1$H) × 220 ($t_1$, $^{15}$N) complex points.

**MS for the determination of alpha-synuclein–Ca complexes.** Samples of 10 µM wild-type alpha-synuclein in Tris buffer (20 mM, pH 7.4) were diluted in 50% methanol/50% dH$_2$O (v/v) to a final concentration of 2 µM. Samples containing 3.6 mM CaCl$_2$ were also prepared as described below. Stock solution of CaCl$_2$ was added to the alpha-synuclein sample and gently mixed with a micropipette, and then incubated for 15 min at room temperature. The sample was then diluted with 50% methanol/50% dH$_2$O (v/v) to a final protein concentration of 2 µM before MS analysis. The effect of formic acid concentration on the MS signal was also investigated: for that reason, the previous samples were also prepared with formic acid to final concentrations of 0.01, 0.1 & 1% (v/v). Samples were infused into a Synapt G2-Si mass spectrometer (Waters, USA) using a syringe pump (CorSolutions, USA) at a flow rate of 3.5 µL/min. Source temperature 80 °C, cone voltage 30 V, desolvation temperature 250 °C, trap collision energy 4 V, transfer collision energy 4 V, Source pressure $7.7 \times 10^{-6}$ bar, Trap pressure $8.8 \times 10^{-6}$ bar, IMS cell pressure $2.6 \times 10^{-7}$ bar, Transfer pressure $8.7 \times 10^{-6}$ bar. All data were collected in positive ion mode.

**Immunofluorescence.** Blocking and permeabilization were performed using 5% serum and 0.01% digitonin in phosphate-buffered saline (PBS) for 1 h. Primary antibodies were incubated for 1 h, followed by four washes with PBS. Secondary antibodies were incubated for 10 min, followed by four washes with PBS. For staining of synaptosomes no digitonin was used, instead all solutions contained 0.05% Tween-20. Samples were kept in PBS containing 5 mM sodium azide (Sigma-Aldrich).

For STED imaging of synaptic vesicles, two primary antibodies, targeting the two most abundant SV proteins[13] synaptophysin (101002, 1:750 dilution, SYnaptic SYstems, Goettingen, Germany) and VAMP2 (104202, 1:750 dilution, SYnaptic SYstems) were used in purpose of improved signal to noise ratio. A secondary anti-rabbit antibody conjugated with ATTO-647N (40839, 1:100 dilution, Sigma-Aldrich) was used to detect both primary antibodies simultaneously. For combined confocal/STED imaging of synaptic vesicles and alpha-synuclein, vesicles were stained with a primary antibody for synaptotagmin-1 (105103, 1:500 dilution, SYnaptic SYstems) and a secondary anti-rabbit antibody conjugated with Alexa-488 (18772, 1:100 dilution, Sigma-Aldrich).

Synaptosomes were stained for alpha-synuclein (D37A6 XP®, 1:500 dilution, rabbit, Cell Signalling, Danvers, US) and VAMP2 (104211, 1:500 dilution, mouse, SYnaptic SYstems, Goettingen, Germany). As secondary antibodies anti-rabbit Alexa Fluor®647 (ab150067, 1:200 dilution, Abcam) and anti-mouse Alexa Fluor®568 (ab175700, 1:200 dilution, Abcam) were used.

VM neurons were stained for alpha-synuclein (ab6162, 1:300 dilution, sheep, Abcam) and synaptotagmin-1 (105103, 1:500 dilution, rabbit, SYnaptic SYstems). As secondary antibodies anti-sheep Alexa Fluor®647 (A21448, 1:200 dilution, Life Technologies) and anti-rabbit Alexa Fluor®568 (A11036, 1:1000 dilution, Life

Technologies) were used. Postfixation was performed with 4% formaldehyde (Sigma-Aldrich) for 10 min to minimize the occurrence of detached fluorophore which would interfere with *d*STORM imaging.

**TEM, STED, and *d*STORM**. For transmission electron microscopy (TEM) imaging of synaptic vesicles 1 µL of SVs (3.3 mg/mL) were incubated in 50 µL of vesicle buffer at 37 °C with and without 50 µM monomeric alpha-synuclein for 4 days without shaking. Overall, 10 µL of each sample was incubated on glow-discharged carbon coated copper grids for 1 min before washing twice with dH₂O. 2% uranyl acetate was used to negatively stain the samples for 30 s before imaging on the Tecnai G2 80-200kv TEM at the Cambridge Advanced Imaging Centre.

For stimulated emission depletion (STED) imaging 0.5 µL of SVs (3.3 mg/mL) in 100 µL vesicle buffer were incubated with either 200 µM CaCl₂ or 1 mM EGTA, with or without 50 µM WT unlabeled alpha-synuclein. The mixture was taken up and down a 30 G needle to disperse the SV, before incubating at 37 °C for 24 h using Lo-Bind Protein Eppendorf tubes. Eight well glass bottom µ-slides (ibidi GmbH, Munich, Germany) were coated with Biotin-PEG-cholesterol according to the protocol described previously[64] and synaptic vesicles were allowed to adhere for 1 h at RT. SVs were fixed with 4% formaldehyde (Sigma-Aldrich) in PBS for 30 min and washed three times with PBS, staining was performed as described above. For combined confocal/STED imaging, 50 µM human WT alpha-synuclein was complemented with 10% of alpha-synuclein N122C mutant labeled with ATTO-647N dye (05316, Sigma-Aldrich). This allowed direct imaging of alpha-synuclein, whereas synaptic vesicles were immunolabelled as described above. STED imaging was performed on a home-built pulsed STED microscope[12]. STED excitation ($\lambda_{exc}$ = 640 nm) and depletion ($\lambda_{depl}$ = 765 nm) were generated from the same titanium sapphire oscillator operating at 765 nm. The beam was divided between two paths. In the excitation path, a supercontinuum was generated by pumping a photonic crystal fiber (SCG800, NKT photonics, Cologne, Germany) and the excitation wavelength was selected by a bandpass filter (637/7 BrightLine HC, Semrock, NY, USA). Excitation and depletion pulse lengths were stretched to 56 and 100 ps respectively through propagation in SF66 glass and polarization maintaining single mode fibers. The depletion beam was spatially shaped into a vortex beam by a spatial light modulator (X10468 02, Hamamatsu Photonics, Hamamatsu City, Japan) and the beams were recombined using a spatial light modulator. Imaging was performed using a commercial point scanning microscope (Abberior Instruments, Göttingen, Germany) comprising the microscope frame (IX83, Olympus, Shinjiuku, Japan), a set of galvanometer mirrors (Quad scanner, Abberior Instruments) and a detection unit. A ×100/1.4 NA oil immersion objective (UPLSAPO 100XO, Olympus) and the Inspector software was used for data acquisition (Andreas Schönle, Max Planck Institute for Biophysical Chemistry, Göttingen, Germany). Fluorescence emission was descanned, focused onto a pinhole and detected using an avalanche photodiode (SPCM-AQRH, Excelitas Technologies. Waltham, USA). A field of view of 20 × 20 µm² and 20 nm pixel size was used. Confocal images of ATTO488 labeled vesicles were correlated with STED images. Images were acquired on the same system as described above. Fluorescence excitation at 488 nm (Cobolt 06-MLD, Cobolt, Solna, Sweden). Fluorescence emission was filtered by a dichroic mirror (ZT594rdc, Chroma, Olching, Germany) and a bandpass filter (FF01-550/88-25, Semrock). 6 images were analyzed per condition for three experiments.

Direct stochastic optical reconstruction microscopy (*d*STORM) was performed as previously described[65]. Briefly, imaging was performed on a Nikon TE inverted microscope using a 100 ×, 1.49 NA TIRF objective lens (Nikon UK Ltd.). Excitation at 640 nm (Toptica Photonic AG, Graefelfing, Germany) was used for Alexa Fluor®647 and 561 nm (Oxxius SLIM-561) for Alexa Fluor®568. Laser beams were collimated and combined by dichroic mirrors and expanded to illuminate the sample for widefield fluorescence microscopy. A 405 nm (120 mW) (Mitsubishi Electronics Corp., Tokyo, Japan) laser was used as reactivation source. The fluorescence light in the detection path was separated from the illumination light by a dichroic filter cube (Semrock multi-edge dichroic Di01-R405/488/561/635-25×36 followed by a FF01-446/523/600/677-25 filter, Semrock, Rochester NY, USA) and was subsequently filtered further using additional bandpass filters (Semrock BP-607/35-25 or BP-642/35-25 for Alexa Fluor®647 and Alexa Fluor®568 respectively). An electron-multiplying charge-coupled device (EM-CCD) camera (Andor iXon DV887 ECS-BV, Andor, Belfast, Northern Ireland) was used for detection. The excitation intensity was 2 kW/cm² for the 640 nm laser and 5 kW/cm² for the 561 nm laser. Single-molecule photoswitching of the Alexa Fluor® 647 and Alexa Fluor® 568 were performed in thiol-containing switching buffer[66] using 100 mM mercaptoethylamine (MEA) in PBS, adjusted to pH 10 using 1 M KOH. Imaging was performed in highly inclined (HiLo) illumination mode[67]. Overall, 16,000 image frames with exposure times between 7.18 and 20 ms were recorded and subsequently reconstructed using either rapidSTORM 3.3[68] or the open-source rainSTORM software developed in-house[69] written in MATLAB (The MathWork Inc., Natick, USA). Image overlays were assembled in FIJI[70], experiments described were repeated at least three times and all images were processed as described in the following section.

**Image analysis**. Western blots were analyzed in FIJI[70]. For the analysis of synaptosomal alpha-synuclein and VAMP2 distribution, individual synaptosomes and their associated fluorophore localizations as found by rapidSTORM were

segmented from the aligned overlay-images using a custom-written Matlab script. From the data files, the size for alpha-synuclein distribution and VAMP2 distribution per synaptosomes were measured using a cluster analysis algorithm with Ripley's K function as cluster size metric[71]. In short, the algorithm measures the Euclidian distances between each localization with every other one. A histogram of the distances is then generated and normalized by a histogram as expected from randomly placed localizations at the same mean density. The maximum distance value of the normalized histogram then yields a measure for the radius of the localization clusters. For image analysis of synaptic vesicles, the spot detector plugin of the image analysis software ICY was used to identify vesicles[72,73], followed by analysis using an in-house MATLAB script to calculate the number of vesicles within a 250 nm radius (see code for image analysis of synaptic vesicle STED images in Supplementary Methods). Analysis of alpha-synuclein and synaptotagmin-1 in ventral midbrain neurons was performed using the particle analysis plugin of the image analysis software FIJI[70]. For co-localization analysis, the Coloc 2 plugin was used[74], based on Pearson's correlation analysis.

**ThT assay and SEC-HPLC quantification of alpha-synuclein monomer**. The aggregation of alpha-synuclein was measured by Thioflavin T (ThT) assay. Briefly, 10 µM ThT was incubated with 100 µL of 100 µM alpha-synuclein with 2.5 mM CaCl₂ or 1 mM EGTA with and without 1 µL 3.3 mg/mL SV. Assays were performed in non-binding, clear bottom, black 96-well plates (PN 655906 Greiner Bio-One GmbH, Essen, Germany) which were sealed with an Ampliseal transparent microplate sealer (Greiner Bio-One GmbH). Plates were incubated with orbital shaking at 300 rpm at 37 °C and the readings of ThT fluorescence intensity at 486 nm were collected every 16 min for 300 cycles in the top excitation/emission mode at a focal height of 5.5 mm. Excitation was set at 440 nm with two flashes using 10% of the excitation light (Envision 2104 Multilabel Reader, PerkinElmer, Turku, Finland). Experiments were repeated three times with six replicates for each condition. The lag time was calculated by extension of a linear fit to the elongation phase, passing through the baseline using the equation: $y = a + b * x$. $k_1$ (nucleation rate constant) and $k_2$ (growth rate constant) were calculated from the Finke–Watzky two-step model fitted to sigmoidal curves of ThT fluorescence[25,75] using the equation below:

$$[B]_t = [A]_0 - \frac{\frac{k_1}{k_2} + [A]_0}{1 + \frac{k_1}{k_2[A]_0}\exp(k_1 + k_2[A]_0 t)},$$

where $[B]_t$ is the concentration of product, $[A]_0$ is the concentration of reactant at time 0 h, $k_1$ is the nucleation rate constant and $k_2$ the elongation rate constant.

10 µL samples of alpha-synuclein fibrils were taken from each condition of the ThT assay and imaged by TEM as described above. Then, samples were removed from the wells and centrifuged at $21,100 \times g$ for 1 h at RT to pellet fibrils and oligomers. The supernatant was removed for quantitative analysis by size exclusion chromatography (SEC). SEC analysis was performed on Agilent 1260 Infinity HPLC system (Agilent Technologies LDA UK Limited, Stockport, UK) equipped with an autosampler and a diode-array detector using BioSep-SEC-2000s column (Phenomenex, Macclesfield, UK) in phosphate-buffered saline (Gibco® PBS, Thermo Fischer Scientific) at 1 mL/min flow rate. The elution profile was monitored by UV absorption at 220 and 280 nm. Remaining monomer concentration of alpha-synuclein was calculated from a calibration curve. Remaining monomer concentration for samples with SVs could not be calculated due to the presence of contaminating SV proteins obscuring the monomeric alpha-synuclein chromatographic profile.

**Statistics**. Statistical analysis was performed using GraphPad Prism 6.07 (GraphPad Software, Inc., La Jolla, CA, USA). Values are given as mean ± s.e.m. unless otherwise stated. Either two-tailed *t*-test or one-way ANOVA with Tukey's post-hoc correction were used as indicated. Significance was considered at $p < 0.05$.

**Data availability**. All relevant data are available from the authors.

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

## Acknowledgements

We would like to thank Dr. Eugene Mosharov and Prof. David Sulzer for fruitful discussions. J.L. was supported by a research fellowship from the Deutsche Forschungsgemeinschaft (DFG; award LA 3609/2-1). M.Z. acknowledges funding from the Eugenides Foundation. C.F.K. acknowledges funding from the UK Engineering and Physical Sciences Research Council (EPSRC). A.D.S. acknowledges funding from the UK Medical Research Council (MRC, MR/N000676/1). A.D.S. and G.F. acknowledge funding from Parkinson's UK (G-1508). G.S.K.S. and C.F.K. acknowledge funding from the Wellcome Trust, the UK Medical Research Council (MRC), Alzheimer Research UK (ARUK), and Infinitus China Ltd. J.L. and A.D.S. acknowledge Alzheimer Research UK (ARUK) travel grants.

## Author contributions

G.F. and A.D.S. performed NMR studies. M.Z. and J.J.P. performed the mass spectrometry studies. A.D.S. and N.N. performed the in vitro studies. A.D.S. and N.C. performed studies on isolated synaptic vesicles. J.L., F.S., A.D.S., and A.T. contributed to synaptosome studies. J.L., F.S., C.H.M., and D.P. contributed to cell studies. J.L., A.D.S, F.S., E.R., M.F., and R.L. were involved in data analysis. W.Z., A.T., P.F., P.St.G.-H., C.F.K., and G.S.K.S. conceived and designed the experiments. J.L., A.D.S., C.H.M., and G.S.K.S. conducted the overall manuscript.

## Additional information

**Competing interests:** The authors declare no competing financial interests.

