## [Peer Review File · Nature Communications]

Reviewers' comments:

Reviewer #1 (Remarks to the Author):

In the present work, the authors investigated the binding properties of the C-terminus of α -synuclein (α S) to lipids. They estimated the affinity of calcium binding to α S, tested whether the binding of calcium by α S differ in the absence and presence of SVs, and studied the interaction of α S and SVs with and without calcium. They used NMR, (CEST) NMR, mass spectrometry, immunofluorescence and superresolution microscopy techniques, and used isolated synaptic vesicles, synaptosomes, and dopaminergic neurons in culture.

Their main findings are that the calcium affinity for α S is 21 μ M, what is within the range of physiological intracellular calcium transients, and that calcium increases the binding of α S to lipids and SVs. They propose that α S is a modulator of SVs homeostasis and that the clustering of synaptic vesicles is regulated by calcium binding to α S.

The manuscript is clear and the experiments are well designed and performed. The results could be of interest for understanding the physiological and pathological properties of α S.

Major concerns:

Of interest is that the Ca-dependent clustering of α S was reversed in the presence of a calcium chelator (EGTA), while VAMP2 distribution was not affected by calcium (Fig. 2C & D). Were 2.5 mM calcium or 1 mM EGTA present at the extracellular medium? In this case, is the synaptosome's membrane permeable to EGTA? Did you use EGTA-AM? How was calcium entry stimulated?

In experiments illustrated in Figure 3, superresolution of isolated synaptic vesicles (SVs), the calcium concentration used was far away from physiological cytosolic values. Do you have the same effect with 100-200 μ M calcium?

Finally, I am a little confused after having compared the present conclusions with those in a previous paper of the authors in the same journal (Fusco et al., 2016. Nat Com). In the 2016's paper, they also proposed that α S induces the clustering of SVs by a double-anchor mechanism consisting in the binding of its N-terminus to one SV and its lipophilic central segment (NAC region) to another SV. Here, however, it seems that the N- and the C-terminus are the binding regions. Is the discrepancy methodological (synthetic lipid vesicles versus synaptic vesicles)? And if not, should be concluded that all three main regions of α S can physiologically bind to lipids? This point should be clarified in the Discussion.

Minor point:

In the graph axis of several figures, it is confusing to see the '/' symbol before the units. For example, in Fig. 1B : $[Ca^{2+}]/mM$, instead of $[Ca^{2+}]$, mM.

Reviewer #2 (Remarks to the Author):

In this study Schierle and collaborators describe a novel calcium-binding feature of alpha-synuclein. They suggest that calcium binds to the C-terminus of alpha-synuclein, and that this region of the molecule also binds to synaptic vesicles. Finally, they suggest that this interaction is involved in the regulation of the localization of alpha-synuclein, in the regulation of synaptic vesicle clustering, and possibly also in synucleinopathies.

As the interaction of alpha-synuclein with synaptic vesicles and other synaptic elements has been analyzed in the past in detail, as the authors acknowledge, the current manuscript needs to be highly convincing, in order to be considered a useful addition to the field. A number of issues, however, are still open, and therefore render the manuscript unsuitable for publication at this time.

Here are the major points:

- 1) The binding of calcium in vitro seems to require millimolar concentrations. Is this the case? If so, the interaction described here has limited relevance, as such concentrations will only be encountered in neurons with a severely damaged membrane, in which physiological processes would be shut off.
- 2) In figure 1d, the effect of calcium does not appear to be significantly different from that of magnesium. The blot shown is very different from the quantification shown, and is therefore misleading.
- 3) Figure 2c contradicts strongly the main claims of the manuscript. It shows that alpha-synuclein colocalizes very poorly with the synaptic vesicle marker VAMP2. In fact, it seems to avoid the synaptic vesicles in these images, which implies that the vesicle-binding reaction that the authors describe in vitro is not particularly strong in real synapses. Also, the localization of synuclein seems to be affected by calcium, while that of the vesicles is not. This again suggests that the interaction described here is not particularly strong, and/or that it requires extremely high calcium levels, which are not encountered in living synapses.
- 4) In Figure 3, alpha-synuclein interacts with synaptic vesicles in vitro, and causes their clustering in the absence of calcium. The clustering actually appears to be reduced in presence of calcium, at least according to the typical images provided, which again contradicts the main statements of the authors.
- 5) The neuron experiments presented in Figure 4 do not seem to add substantially to the manuscript. The effects of a calcium channel blocker are extremely complex in this context, since calcium entry has a wide range of effects, in addition to the potential effects on alpha-synuclein.
- 6) The effects presented in Figure 5 are not convincing, especially as the experiment has only been repeated twice. The fact that no vesicles are actually seen in the preparation made in presence of synaptic vesicles is confusing.

Reviewer #3 (Remarks to the Author):

The central question addressed by this manuscript is how does calcium modulate the physiological function of alpha-synuclein? The authors show that in the presence of Ca²⁺

synuclein binds isolated synaptic vesicles not only through its N-terminal region, as previously shown, but also through its C-terminus. The latter interaction is modulated by Ca^{2+} binding to synuclein and is a key determinant of synaptic vesicle clustering.

This manuscript represents a major advancement relative to the previous model of synuclein – lipid vesicle interactions (i.e. the “double anchor mechanism”) and one of the major strengths of this manuscript is that it focuses on physiological synaptic vesicles isolated from rat brains as opposed to the simpler lipid vesicles used in previous studies.

Considering that large calcium fluctuations occur *in vivo* at the pre-synapsis during neuron depolarization, this manuscript is highly relevant for the physiology and pathology of synuclein *in vivo*. Another notable strength of this ms is the excellent integration of orthogonal experimental techniques (NMR, MS, immunofluorescence, TEM, STED, dSTORM, Tht and SEC assays).

Minor points:

- The authors mention that the binding of calcium to synuclein was analyzed in terms of a “multiple ligand model”. This term is vague and should be explained better. For example, is this multiple ligand model a Scatchard model where all calcium binding sites are equivalent and independent of each other? Or is this model a Hill-like model in which calcium binding occurs without intermediates in a highly cooperative manner? The latter scenario seems more consistent with the 2nd equation in p. 34, however in this case the concentration of free calcium in the 1st equation in p.34 should appear at the power of “L”. It is possible that the effective affinity may not change after these ambiguities about the model are clarified, but this needs to be addressed so that the first two equations of p. 34 are reconciled.
- What are the units for CS in Fig. Supp 1. Ppm? If so these CS changes seem larger than those in Fig. 1. Pls clarify.
- Fig. 2a: We suggest including a +/- calcium difference plot to show more clearly the effect of Ca^{2+} . Alternatively, a smoothed average trend line could be superimposed to the current DEST plots to better show the variations induced by calcium.
- In page 8, the endogenous levels of alpha-synuclein should be specified to let the reader appreciate the concentration difference relative to what is called high synuclein concentration (i.e. 50 μM)
- P. 35: was the reference offset for CEST only at -100 kHz or at both +/-100 kHz?
- In p. 41, the exp is missing the last parenthesis “)”

Reviewer #1

We would like to thank reviewer 1 for his/her very constructive criticism. There were three major points the reviewer wanted us to address:

(i) Clarification of the experimental set-up described in Figure 2c&d

Both reviewer 1 and 2 commented on this and we apologize that this has not been clear. We provided more details on the experimental procedures in the text and added new experiments to make this point clearer.

(ii) Does synaptic vesicle clustering also occur at lower, physiological calcium concentrations?

This is a very valid point and also reviewer 2 wanted this point to be addressed. We thus performed the experiments described in Fig. 3 at lower, physiological calcium concentrations and show that synaptic vesicle clustering already occurs at much lower calcium concentrations than shown before.

(iii) How do our current results compare to our previous results published in 2016? (*Fusco et al. 2016 Nat Commun*)

We agree with the reviewer that this point has not been very clear and we have now significantly amended the text to clarify this.

Point-to-Point discussion

Point1

- The reviewer asked to specify the conditions used for the incubation of synaptosomes (Figure 2c and d). We thank for this comment, as it helped us to make this point clearer in the manuscript.

The synaptosomes were either incubated with an extracellular solution containing 2.5 mM calcium which represents normal physiological conditions (i.e. low calcium concentrations in the intracellular space and high calcium concentrations in the extracellular space) or with an extracellular solution containing 1 mM EGTA. We clarified this point in the text pointing out that the alpha-synuclein clustering is seen under physiological conditions (page 7). Furthermore, we termed the EGTA condition "calcium starved/depleted", as we think this probably describes the treatment better, as the reviewer is right we did not directly add EGTA into the synaptosomes (via permeabilization or using EGTA-AM) for the reasons that direct intracellular manipulation could affect other pathways in intact synaptosomes.

- In addition to this, the reviewer asked how we had increased the calcium concentration in the intracellular space.

Since the experiments we showed were done under normal physiological calcium conditions we have now added a new set of experiments for which we have increased the intracellular calcium concentration by stimulating the synaptosomes with 70 mM KCl (affecting membrane potential and leading to calcium entry via voltage-gated calcium channels). We did not observe a further change in the distribution of alpha-synuclein (cluster radii did not differ significantly). We included these experiments in the supplement, Supplementary figure 6 and clarified these points in the text, page 7/8.

Point 2

- The reviewer questioned the physiological relevance of our in vitro experiments on synaptic vesicle clustering.

As the reviewer suggested, we performed additional experiments using 200 μ M calcium rather than 2.5 mM calcium to show that the clustering is already relevant at physiological calcium concentrations such as occur during synaptic stimulation (up to several 100 μ M, Llinás et al., 1992; Schneggenburger and Neher, 2000) (Figure 3a, page 8/9). We feel that this experiment has added further strength to our study as 200 μ M relates better to the KD we have found for alpha-synuclein and calcium interaction in vitro using NMR (20 μ M).

Point 3

- The reviewer wanted us to clarify how our current data compare with the data published in Fusco et al Nat Comm 2016. We agree that this point was not very clear and we have now discussed this in more detail which we think will add further value to the manuscript.

In a previous study we have shown that the modes of binding of the N-terminal and NAC-region of alpha-synuclein to small unilamellar vesicles (SUVs) of DOPE:DOPS:DOPC (in 5:3:2 molar ratios) are independent of each other. This degree of independence suggested that the two regions can bind the same vesicle but also multiple vesicles. We referred to this N-terminus and NAC-region binding of alpha-synuclein to SUVs as double anchor mechanism. In our current study we have characterised the binding of alpha-synuclein to synaptic vesicles isolated from rat brain using CEST NMR, showing again that the strongest binding occurs at the N-terminus and revealing an intermediate level of synaptic vesicle interaction of the NAC-region and the C-terminus. Upon addition of calcium this binding via the C-terminus, and also for certain residues of the NAC-region, was found to be increased. We refer to this now as extended double anchor mechanism since the double anchor has been extended to the C-terminus of alpha-synuclein.

We acknowledge that the first version of the paper was not entirely clear on this aspect and have revised the manuscript accordingly (page 6/7 and 12/13).

Point 4

- Axes labelling is confusing.

We agree with the reviewer and have changed the axes labelling in Figure 1b and Figure 4, putting the units into brackets like in all other figures.

Reviewer #2

We would like to thank the reviewer for his/her very constructive criticism. The major points from reviewer #2 refer to the physiological relevance of our findings and the repeatability of our results. To address these points we

- (i) included experiments on synaptic vesicle clustering at 200 μM calcium, a concentration within the range of synaptically stimulated neurons
- (ii) repeated the Western blot of alpha-synuclein lipid binding
- (iii) included new experiments on calcium stimulated synaptosomes and clarified the experimental set up in the text
- (iv) added an extra repeat to the ThT assay and included TEM analysis of alpha-synuclein synaptic vesicle samples at the start of the aggregation experiment

Point-to-Point discussion

Point 1

- Reviewer 2 wondered whether our results in vitro are physiologically relevant as they require mM concentrations.

Our NMR-based calcium titration experiment determined an alpha-synuclein calcium binding constant of 21 μM , which is well within the range of physiological concentrations such as occur during synaptic stimulation (up to several 100 μM , Llinás et al., 1992; Schneggenburger and Neher, 2000). For NMR studies high protein concentrations are required (200 μM), thus higher concentrations of calcium had to be used for stoichiometric reasons. The titration was done up to 18 equivalents of [alpha-synuclein] which are ranges of ligand commonly found in titrations of other calcium binding proteins using NMR (Prebil et al., 2016; Ubach et al., 1998).

However, for the in vitro data described in Fig. 3 we had used millimolar calcium concentrations. This was also a concern that was raised by reviewer 1 (see comment to reviewer 1 point 2). We thus performed new experiments and studied synaptic vesicle clustering at 200 μM rather than at 2.5 mM calcium. We now show that

synaptic vesicle clustering already occurs at physiological calcium concentrations (Figure 3a, page 8/9).

synaptic vesicle clustering already occurs at physiological calcium concentrations reached during synaptic activity (Figure 3a, page 8/9).

Point 2

- The reviewer asked for the quantification of the Western blot shown in Figure 1d.

We agree that the Western blot shown in Figure 1d did not represent the statistics shown. We have now repeated the experiments and show a more representative image in Fig. 1d. For further transparency, we also added our Western blot repeats to the supplement (Supplementary figure 3).

Point 3

- The reviewer asked on the interpretation of our results on synaptosomes regarding the distribution of alpha-synuclein and VAMP2 shown in Fig. 2c&d. In particular, the reviewer thinks that the vesicle-binding reaction that we describe in vitro is not particularly strong in real synapses. We apologize and we hope our new data and our clarifications in the text will help to understand our results better.

We have added new data to show that synaptic vesicle clustering already takes place at 200 μ M calcium which is well within a physiological range (see also comment to point 1 above). Furthermore, our synaptosome experiments were done under normal physiological conditions (normal physiological calcium in the intracellular space and high extracellular calcium). We have now clarified this in the text and apologize if this has not been clear. In order to address whether increased intracellular calcium concentrations have a further effect on synaptic vesicle clustering in synaptosomes we added new experimental data for which we stimulated synaptosomes with 70 mM KCl (affecting membrane potential and leading to calcium entry via voltage-gated calcium channels; see also point 1 of reviewer1). We did not observe any further clustering of alpha-synuclein confirming that already normal physiological calcium concentrations can lead to alpha-synuclein clustering.

The reviewer is correct, that in (Figure 2c), under physiological conditions (normal intracellular calcium and high (2.5 mM) extracellular calcium), we see that alpha-synuclein staining does not overlap with VAMP2 staining. We agree that this could be interpreted as if alpha-synuclein does not bind to synaptic vesicles at all. We do, however, not believe that this is the case for two main reasons: First, all our in vitro data as well as data from others (Georgieva et al. 2008, Jao et al. 2008, Bodner et al. 2009, Middleton et al. 2010, Trexler et al. 2010, Ulmer et al. 2005) show strong binding of alpha-synuclein to synaptic vesicles. Second, we have preliminary evidence that alpha-synuclein binds to only a subset of synaptic vesicles (similar to what has been

indicated in Lee et al. 2008 Acta Neurobiol Exp). We have now added a new section in the discussion to make this point clearer (page 14). In order to fully determine which synaptic vesicles are positive for which synaptic vesicle marker requires significantly more experiments which we feel are beyond the scope of the current manuscript.

Point 4

- Reviewer 2 and reviewer 1 raised the question about the physiological relevance of our in-vitro experiments shown in Fig 3. As suggested, we have performed the synaptic vesicle experiment at 200 μ M calcium, which confirms that this effect is physiological relevant.

We now show that 200 μ M calcium already leads to synaptic vesicle clustering (see points 1 and 3 above). In particular, we see that there is a higher amount of single synaptic vesicles in the EGTA group compared to the other 3 groups, while clusters of 2 vesicles are increased for incubations in the presence of calcium, asyn + calcium and also of asyn + EGTA compared to EGTA alone, which is consistent with our previous experiment at higher calcium concentrations. While the values for asyn + EGTA indicate less clustering compared to alpha-synuclein incubation in the presence of calcium, they are not significantly different. We believe that this is due to both alpha-synuclein and calcium being able to cluster synaptic vesicles as we have shown in our previous paper (Fusco et al. 2016).

Point 5

- The reviewer indicated that the experiments done with isradipine on neurons do not add substantially to the manuscript.

We disagree with the reviewer on this point. Isradipine is a voltage-gated calcium channel blocker and thus reduces the amount of calcium that can enter upon neuronal stimulation such as upon dopamine treatment. We show that synaptic vesicle clustering occurs upon dopamine treatment which is blocked by isradipine and we also show that toxicity is blocked by the addition of isradipine. We thus provide further evidence that controlling the calcium levels, such as by using a voltage gated calcium channel blocker, may be a protective strategy. We thus feel that these experiments are relevant and support our hypothesis that increased calcium concentrations may enhance toxicity.

Point 6

- The reviewer pointed out that the experiment in Figure 5 was only repeated twice and that synaptic vesicles should be visible in the TEM images shown in Fig.5c.

We added another repeat to the ThT assay and now show the combined results. The repeat shows again that there is a strong effect of calcium and a small effect of synaptic vesicles on the aggregation propensity of alpha-synuclein under the conditions we used. The reviewer is right, we have never seen any vesicular structures at the end of our aggregation assay (after 7 days of incubation). In order to verify that synaptic vesicles are present during the aggregation assay we now show a TEM image of monomeric alpha-synuclein in the presence of synaptic vesicles at the start of the aggregation experiment (Supplementary figure 10). We added a paragraph to the manuscript (page 11), outlining that we do not see synaptic vesicles present at the end of the aggregation reaction and make reference to the supplementary figure.

Reviewer #3

We would like to thank reviewer 3 for highlighting the advancement of our approach to previous approaches and for mentioning that our manuscript is highly relevant for the physiology and pathology of synuclein. There were three main points reviewer #3 wanted us to address:

(i) Explain “multiple ligand model”

We addressed this in the text and added the respective reference.

(ii) Define the units for chemical shifts in Figure 1a and supplementary Figure 1

We corrected and defined this in the text.

(iii) Improve Figure 2a

As suggested we added a smoothed average trend line to Fig. 2a.

Point-to-Point discussion

Point 1

- Define the multiple ligand model. We agree with the reviewer and we have now provided more detail on our choice of the multiple ligand model.

We used a model that has been previously used to characterize the binding of alpha synuclein to lipids (Galvagnion et al Nat Chem Bio, 2015). The advantage of this model is that it is based on the total concentration of ligands, which we know accurately, whereas other models such as the Hill equation for multiple binding sites requires the knowledge of the free ligand concentration, which is quite challenging to measure since calcium is in fast exchange with alpha-synuclein. In particular, as in our case the concentration of the protein is close to the KD we cannot approximate

free [L] with total [L]. We have now provided a more detailed set of equations to explain the basis of our model and added a reference for further details (Galvagnion et al Nat Chem Bio, 2015) (see page 34/35 and 5).

Point 2

- The referee pointed out that there are discrepancies between Figure 1a and Supplementary figure 1, which we addressed accordingly.

We thank the referee for this point and have now corrected the legend of figure 1a. We also confirm that the units of Fig. S1 are in ppm (these have now been added to the plot). The peak changes in figure 1a are not as high as the maximum changes in the supplementary figure 1 (3.6mM calcium) as Figure 1a shows an intermediate point in the titration (at a calcium concentration of 1.6 mM). We have amended the legend of figure 1a to clarify this.

Point 3

- We thank the reviewer for the comment on figure 2a as the new figure helps the reader to see the differences between the two measurements better.

We included the CEST Figure 2a with a smoothed average trend line.

Point 4

- The reviewer wanted us to compare the alpha-synuclein concentration used for the experiment shown in Figure 3a with the endogenous alpha-synuclein concentration.

The value estimated for the endogenous presynaptic alpha-synuclein concentration is 43 μ M (Takamori et al., 2006), thus the concentration used for our experiment will be equivalent to twice the endogenous concentration (as seen in disease upon gene duplication). We have now added the alpha-synuclein concentration on page 8.

Point 5

- The reviewer asked about the offset for the CEST measurements.

We employed -100 kHz to measure the reference in the CEST experiment as for our previous experiments published by Fusco et al. 2016.

Point 6

- Fill in missing “)”.

We corrected the missing “)”.

REVIEWERS' COMMENTS:

Reviewer #1 (Remarks to the Author):

The quality of the ms has increased and the authors have responded adequately to all the comments and questions made by this reviewer.

Please, specify in the text that the 1mM EGTA solution does not have added calcium (see lines 158 and 159).

Reviewer #2 (Remarks to the Author):

The authors have responded to most of my comments. They have quantified some of their elements better, included more experiments where the statistics were not convincing, and generally made an effort to improve the manuscript.

They have not responded convincingly to one issue: their demonstration that alpha-synuclein colocalizes properly with synaptic vesicles in living synapses. I raised this point for the original manuscript, mentioning that the authors' own work shows that the two do not colocalize. This is still largely the case in figure 2c, both under physiological conditions and during calcium depletion. The authors included some comments on this, based on preliminary data (not shown) and on the literature. I suggest that the manuscript be published, but only after the addition of such comments in the section dealing with this figure (on pages 7 or 8 of the manuscript).

Reviewer #3 (Remarks to the Author):

The authors have addressed all our previous questions. The reference to the work of Galvagnion et al Nat Chem Bio, 2015 was useful to clarify the binding model for Ca²⁺. Similar models have also been utilized before to analyze multi-valent interactions with amyloidogenic peptides (doi: 10.1016/j.bpj.2013.08.025). Inclusion of this reference would further strengthen the ms. Similarly, the impact of the discussion of the CEST/DEST experiments would be enhanced by including other examples of comparative ST analyses for other amyloidogenic peptides (doi: 10.1021/jacs.7b05012 and doi: 10.1074/jbc.M117.792853). Minor typo in p. 35, line 814: the square parenthesis should be limited to Ca²⁺ and should not include L in the denominator. Otherwise this is an excellent manuscript.

Point-by-point response to the reviewer comments

Reviewer #1 (Remarks to the Author):

The quality of the ms has increased and the authors have responded adequately to all the comments and questions made by this reviewer.

Please, specify in the text that the 1mM EGTA solution does not have added calcium (see lines 158 and 159).

We stated "omitting calcium" at the respective paragraph (page 7/8)

Reviewer #2 (Remarks to the Author):

The authors have responded to most of my comments. They have quantified some of their elements better, included more experiments where the statistics were not convincing, and generally made an effort to improve the manuscript.

They have not responded convincingly to one issue: their demonstration that alpha-synuclein colocalizes properly with synaptic vesicles in living synapses. I raised this point for the original manuscript, mentioning that the authors' own work shows that the two do not colocalize. This is still largely the case in figure 2c, both under physiological conditions and during calcium depletion. The authors included some comments on this, based on preliminary data (not shown) and on the literature. I suggest that the manuscript be published, but only after the addition of such comments in the section dealing with this figure (on pages 7 or 8 of the manuscript).

A paragraph clarifying the polarization of alpha-synuclein was added, relevant references are cited (page 7).

Reviewer #3 (Remarks to the Author):

The authors have addressed all our previous questions. The reference to the work of Galvagnion et al Nat Chem Bio, 2015 was useful to clarify the binding model for Ca²⁺. Similar models have also been utilized before to analyze multi-valent interactions with amyloidogenic peptides (doi: 10.1016/j.bpj.2013.08.025). Inclusion of this reference would further strengthen the ms. Similarly, the impact of the discussion of the CEST/DEST experiments would be enhanced by including other examples of comparative ST analyses for other amyloidogenic peptides (doi: 10.1021/jacs.7b05012 and doi: 10.1074/jbc.M117.792853). Minor typo in p. 35, line 814: the square parenthesis should be limited to Ca²⁺ and should not include L in the denominator. Otherwise this is an excellent manuscript.

Respective references were added (page 5 and page 22). The typo in the methods section was revised (page 22).